# Tunable multistate field-free switching and ratchet effect by spin-orbit torque in canted ferrimagnetic alloy

Cheng-Hsiang Hsu [1,2] ✉, Miela J. Gross [3], Hannah Calzi Kleidermacher [1], Shehrin Sayed[1,2] & Sayeef Salahuddin[1,2] ✉

Spin-orbit torque is not only a useful probe to study manipulation of magnetic textures and magnetic states at the nanoscale but also it carries great potential for next-generation computing applications. Here we report the observation of rich spin-orbit torque switching phenomena such as field-free switching, multistate switching, memristor behavior and ratchet effect in a single shot, co-sputtered, rare earth-transition metal $Gd_xCo_{100-x}$. Notably such effects have only been observed in antiferromagnet/ferromagnet bi-layer systems previously. We show that these effects can be traced to a large anistropic canting, that can be engineered into the $Gd_xCo_{100-x}$ system. Further, we show that the magnitude of these switching phenomena can be tuned by the canting angle and the in-plane external field. The complex spin-orbit torque switching observed in canted $Gd_xCo_{100-x}$ not only provides a platform for spintronics but also serves as a model system to study the underlying physics of complex magnetic textures and interactions.

Spin-orbit torque (SOT) driven magnetization switching has great potential for next-generation magnetic memory technology[1,2]. At the same time, it serves as a powerful probe to study spin-dependent phenomena such as topological spin textures and quantum materials[3,4]. However, despite substantial effort over the last decade, challenges remain in terms of further lowering the critical switching current density, exploring complimentary metal oxide semiconductor (CMOS) compatible materials with high spin-torque efficiency, engineering field-free switching designs[2], etc. As a result, the search for new materials and new spin/magnetic phenomena remains an ongoing effort[5–11]. In this regard, ferrimagnets have been an emerging material system that possesses abundant magnetic properties for interesting phenomena and potential spintronics applications[12]. Ferrimagnets are composed of two different magnetic sublattices that are anti-ferromagnetically coupled, which yields a non-zero magnetization[13]. As a result, ferrimagnets combine the unique properties of both ferro-magnets (FM) and antiferromagnets (AFM), enabling magnetic properties and phenomena such as bulk perpendicular magnetic

anisotropy (PMA)[14], bulk Dzyaloshinskii-Moriya Interaction (DMI)[15], magnetic skyrmion host[16], spin-orbit torque switching[17], and all-optical switching[18]. While there exist many different families of ferrimagnets, rare earth (RE) - transition metal (TM) ferrimagnetic alloy stands out for spintronics applications[12,14,17,18] due to its highly tunable magnetic, electronic, thermomagnetic, optical properties and relatively simple thin film deposition method[12,14,19,20].

Among many fascinating properties of RE-TM ferrimagnetic alloys, the bulk perpendicular magnetic anisotropy (PMA), which even persists in thick films in the micro-meter regime[14,20] is the driving force behind all the fascinating magnetic phenomena. However, switching a PMA magnet with an in-plane polarized spin current requires an in-plane symmetry-breaking field to achieve deterministic switching[21–23]. Despite many efforts in studying RE-TM ferrimagnetic alloys for spin-tronics applications[12,14–20,24,25], it is not until recently that field-free spin-orbit torque switching can be achieved in RE-TM ferrimagnetic systems[26–29]. It is not surprising that RE-TM ferrimagnets carry the capability of inducing a tilt in its anisotropy since it possess properties

[1]Department of Electrical Engineering and Computer Science, University of California, Berkeley, California, USA. [2]Materials Science Division, Lawrence Berkeley National Laboratory, Berkeley, California, USA. [3]Department of Physics, University of California, Berkeley, California, USA.
✉e-mail: chhsu@berkeley.edu; sayeef@berkeley.edu

of both FM and AFM. In fact, one of the most common systems designed to induce a tilt in its PMA for field-free SOT switching is the AFM/FM bi-layer, where an exchange bias is present[23,30].

In AFM/FM bi-layer systems, not only field-free switching[23,30,31] can be realized but also many interesting magnetic textures and switching behaviors such as electrical control of antiferromagnetic order[32,33], multistate switching[23,34], and memristor behavior[23,34] can be observed owing to its dynamic and intricate magnetic interaction at the AFM/FM interface[35]. Due to the exchange coupling at the AFM/FM interface, AFM spin configuration at the interface can be controlled via SOT and exhibits an exchange spring effect that manifests in a ratchet behavior of the magnetic states[35] and antiferromagnetic states[36,37]. Since ferrimagnetism combines both FM and AFM properties, it is possible to observe not only tilting in its magnetic anisotropy but also possibly all the switching behaviors found in AFM/FM systems, such as field-free switching, multistate switching, memristor behavior, exchange spring effect, and ratchet effect. In RE-TM ferrimagnetic alloys, the parameter space for material design is also much wider with knobs on atomic concentration between RE and TM, thickness scaling (not limited to ultrathin ~ 1 nm thickness due to interfacial PMA), different RE and TM elements, growth condition, and operating temperature. With this motivation, we explore the possibility of achieving the interesting switching behaviors and phenomena observed in AFM/FM systems in a single-layer ferrimagnetic GdCo alloy without any superlattice design that is deposited in a single shot.

In this paper, we report field-free spin-orbit torque switching of GdCo near its magnetic compensation with canted anisotropy and exchange spring behavior at room temperature via transport and magnetometry measurements. Through angle-dependent anomalous Hall resistance field loop measurement, the angle of the canting can be characterized. In addition, multistate switching, memristor behavior, and ratchet effect are observed where the multistate and ratchet effect is tunable with the in-plane symmetry-breaking field. This demonstrates that single-shot RE-TM ferrimagnetic alloys possess similar properties to AFM/FM bi-layers, which expands the material design space for spintronics applications.

## Results

### Thin film deposition and characterization of magnetic anisotropy in GdCo

Canted $Gd_xCo_{100-x}$ are deposited via co-sputtering of elemental Gd and Co targets with base pressure around 1e-8 Torr prior to deposition. The canted $Gd_xCo_{100-x}$ thin films are deposited on the thermally oxidized silicon substrate with 8 nm Ta as the underlayer and capped with 2 nm Pt to prevent oxidation (see Fig. 1a). In this study, three samples (sample-45, sample-56, and sample-13) with different anisotropy canting angles are deposited with the nominal Gd concentrations to be around 24 ~ 25% (see "Methods" Table 1), which is close to the magnetic compensation point[19,20,38].

The thin film magnetometry conducted under room temperature on sample-45 reveals an exchange spring behavior and a large canting of magnetic anisotropy (see Fig. 1b). The magnetization hysteresis loops taken along the OOP and IP directions show comparable remanent magnetization in both directions, indicating a large canting and the total effective anisotropy is close to none. Not only the remanent magnetization is comparable, but also a two-phase switching characterized by a small coercive field and a large coercive field can be observed in both IP and OOP directions. This two-phase switching indicates the $Gd_xCo_{100-x}$ layer is most likely composed of a soft magnetic layer and a hard magnetic layer, which is a signature of exchange-spring behavior[39]. This phenomenon has also been observed in other RE-TM ferrimagnetic systems such as TbCo, especially when the thickness is ultrathin (<10 nm)[39]. In TbCo, the first 2 nm layer is composed of a soft low-density magnetic layer. The sequential layers deposited are much denser and have harder magnetic layers. The coexistence of the soft and hard magnetic layers in

RE-TM ferrimagnetic alloy that are exchange coupled serve as a great exchange spring system to engineer a tunable anisotropy tilting through controlling the concentration, growth condition, underlayer, and overlayer[39,40]. Here, in $Gd_xCo_{100-x}$, this effect can be amplified due to the symmetric filling of the $4f$ shell in Gd compared to Tb (details see Supplementary Note 1).

In order to quantify the anisotropy canting angle of each sample, we present a new way of characterizing anisotropy canting angle through anomalous Hall effect (AHE) measurements with the field direction to be a function of the angle between the z-axis and x-axis (see Fig. 1c inset). Most often, for thin-film magnets, the magnetic easy-axis is either strictly in-plane ($\theta_B = 90°$) due to the shape anisotropy or strictly out-of-plane due to strong interfacial or bulk PMA ($\theta_B = 0°$). In our case, due to the large canting of the magnetic anisotropy, the easy axis will be between 0° and 90°. In a typical AHE measurement, when the field is applied in the magnetic easy-axis, the AHE hysteresis will behave as constant resistance values above the two coercive fields ($B_c$), and the switching between the two constant resistance values occurs upon crossing the coercive fields. In the case of the field being applied in the magnetic hard axis, the AHE hysteresis curve is a sloped line with minimal hysteresis opening below the anisotropy field before the moments are saturated. This is because there are magnetizations in which the easy-axis does not align with the field direction thus, as the field increases, these magnetizations are pulled into the field direction and away from their magnetic easy-axis. However, when the external field direction is aligned with the magnetization easy-axis, the slope in the higher field region (above $|B_c|$) should be close to zero since all moments are saturated. As a result, when the AHE measurements are done with the field applied in a direction between the easy axis and the hard axis, the superposition of the AHE hysteresis from both cases (easy axis and hard axis) occurs, and the AHE resistance should be a sloped line in the region above the coercive fields. It is to be noted that the canting angle obtained through this method represents the canting of the magnetic easy-axis in the net magnetization of the fabricated GdCo heterostructure device. This canting angle is different from the tilting angle of the OOP magnetization often seen in field-free switching literature[26,27] because there is a very minimal IP magnetization component in these systems.

We patterned the thin films into Hall bar devices ("methods") and carried out the angle-dependent AHE measurement on sample-45. In Fig. 1c and Supplementary Fig. S1a, a series of AHE hysteresis loops measured at different field directions $\theta_B$ are shown. In the region above the coercive fields, the slopes of the curve decreases in magnitude as the field direction approaches 45 degrees, and once it crosses 45 degrees, the slope magnitudes begin to increase again (see Fig. 1d). By fitting a line to the region above the coercive fields (see Fig. 1c), we are able to find the angle at which the minimum slope is obtained. In sample-45, the minimum slope is found to be 45 degrees away from the z-axis (see Fig. 1d). This confirms that the magnetic easy-axis in sample-45 is indeed largely canted from the typical IP or OOP directions.

### Recipe for inducing large canting in GdCo ferrimagnet

The canting in GdCo can be induced through oxygen incorporation inside the GdCo layer. To study the effect of oxygen to induce canting, we fabricated Hall bar devices on Ta (8 nm) / $Gd_xCo_{100-x}$ (10 nm) / Pt (2.5 nm) thin film heterostructure that is deposited with composition near the magnetic compensation at room temperature (Fig. 2a). Without any oxidation, the device shows a strong in-plane anisotropy characterized by the sloped straight line in the AHE field loop (Fig. 2b). As we treat the device to a mild oxygen plasma (50 W, 180 mTorr, 50 °C) with a set duration, the shape of the AHE loop transformed from a straight line into a switching loop with improved squareness. After a set amount of oxidation, the squareness no longer improves and rather may degrade if further oxidation is carried out. At this point, we can let the sample stay in an ambient environment for at least a day, the

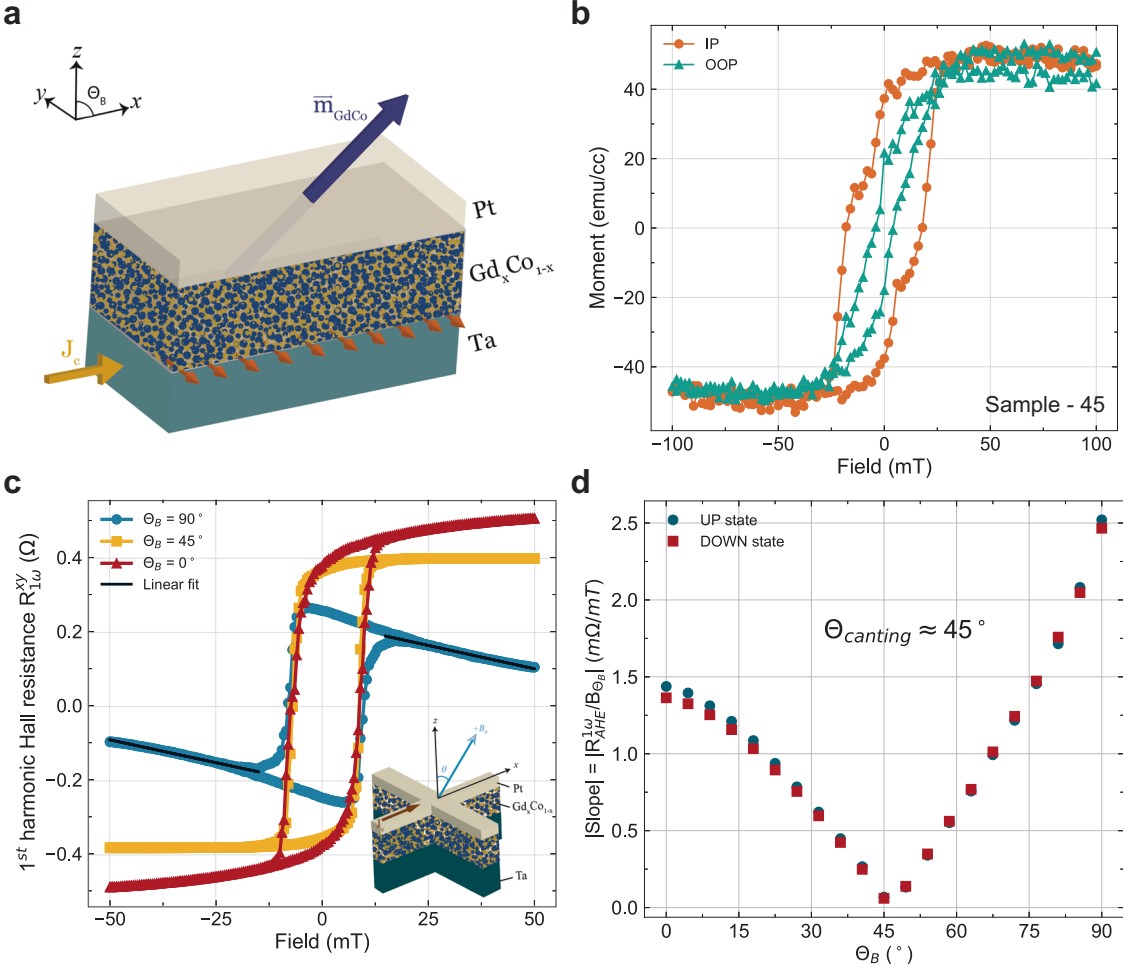

**Fig. 1 | Magnetic characterization of the thin film heterostructures (sample-45) with large canting angles through vibrating sample magnetometer and angle-dependent anomalous Hall field sweep measurement under room temperature. a** The magnetic heterostructure under study is composed of an 8 nm Ta bottom layer as spin Hall source for switching experiments, 10 nm $Gd_xCo_{100-x}$ as the ferrimagnet with canted magnetic anisotropy, and capped with a thin 2 nm Pt. The top Pt layer serves as a protection layer to prevent oxidation of the ferrimagnet and enhances the PMA. **b** Magnetization hysteresis of sample-45 shows that $Gd_{24.7}Co_{75.3}$ is close to magnetic compensation at room temperature, characterized by the small saturation magnetization. By comparing scans done in the IP and OOP direction, the magnetic anisotropy exists in both directions, with the IP anisotropy being slightly stronger, characterized by the remanent magnetization ($M_{r,ip} \approx 20$

emu/cc and $M_{r,oop} \approx 40$ emu/cc) and the larger IP coercive field. Exchange spring behavior can also be observed in both directions, with the IP direction being more pronounced. **c** Linear fit to the field region above the coercive field to obtain the slope in angle-dependent anomalous Hall effect field sweep measurement. The shape of the $R_{AHE}$ vs. $B_{\theta_B}$ can be governed by the angle between the easy axis of the magnetization and the external field direction. By looking at the two extremes ($\theta_B = 0°$ and $90°$), the $R_{AHE}$ is sloped in the field region above the coercive fields ($|B_{\theta_B}| > 10$ mT). For $\theta_B = 45°$, the slope is nearly zero. Inset, angle ($\theta_B$) dependent AHE measurement setup schematics and Hall bar device geometry. $\theta_B$ is defined from the z-axis toward the x-axis. **d** Slope of the high field region as a function of external field angle from the z-axis. With this technique, the canting angles $\theta_B$ are found to be 45° for sample-45, where the slope is the minimum.

## Table 1 | Magnetic properties of canted $Gd_xCo_{100-x}$ moments with different magnetic canting angles in this study

| Magnetic Property | Sample-45 | Sample-56 | Sample-13 |
|---|---|---|---|
| As Deposited Atomic Concentration | $Gd_{24.7}Co_{75.3}$ | $Gd_{24.7}Co_{75.3}$ | $Gd_{24}Co_{76}$ |
| Relative Degree of Oxidation Treatment | Intermediate | Light | Heavy |
| Canting Angle of Easy Axis (∘) | 45 | 56.25 | 13.5 |
| Saturation Magnetization $M_s$ (emu/cc) | 40 | 94 | 111 |
| IP Remanent Magnetization $M_{r,ip}$ (emu/cc) | 37 | 44 | 17 |
| OOP Remanent Magnetization $M_{r,oop}$ (emu/cc) | 20 | 75 | 102 |
| IP Coercive Field $B_{c,ip}$ (G) | 180 | 87 | 49 |
| OOP Coercive Field $B_{c,oop}$ (G) | 41 | 95 | 97 |

squareness then further improves (Fig. 2b). By conducting the angle-dependent AHE field sweep measurement, we can characterize the canting angle of the easy-axis after 1 day and 58 days which turns out to be 22.5 and 13.5 degrees from OOP axis (Fig. 2c, d). We believe the step for the sample to stay in an ambient environment is for oxygen to slowly diffuse into the bulk of the GdCo layer and create an OOP anisotropy (Supplementary Note 2). By controlling the duration of the oxygen plasma step and the duration of the ambient exposure (or acceleration through elevated temperature annealing), a canting angle anywhere between IP and OOP should be obtainable with reasonable optimization.

### Field-free spin-orbit torque switching in canted GdCo heterostructures

With the magnetic anisotropy canting angle quantified, we first performed the spin-orbit torque switching experiments to realize the possibility of SOT switching in the absence of a symmetry-breaking

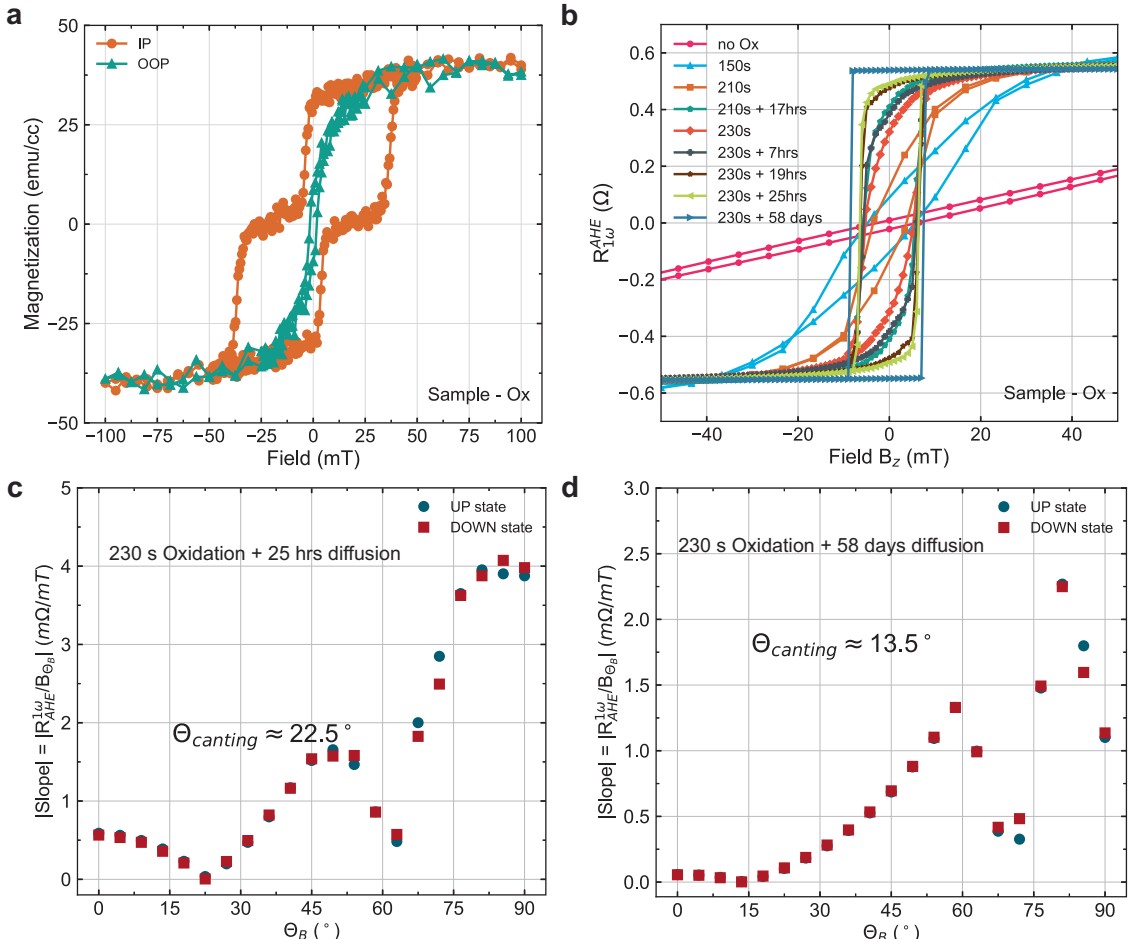

**Fig. 2 | Oxidation of GdCo heterostructure for inducing magnetic canting.**
**a** Magnetization hysteresis of sample-Ox shows that $Gd_xCo_{100-x}$ is close to magnetic compensation at room temperature, characterized by the small saturation magnetization. **b** Anomalous Hall effect hysteresis loop as a function of oxidation duration and ambient exposure duration. **c** Slope of the high field region as a function of external field angle from the $\hat{z}$-axis. With this technique, the canting angles $\theta_B$ are found to be 22.5° for sample-Ox after 230 s of oxygen plasma treatment and 1-day ambient exposure where the slope is the minimum. **d** 13.5° canting is found for sample-Ox after 230 s of oxygen plasma treatment and 58 days ambient exposure where the slope is the minimum.

field on the canted GdCo samples. We performed pulsed current measurements (200 µs) on Hall bar devices (see Fig. 3a) with anomalous Hall effect as the voltage readout mechanism to probe the magnetic state of the canted GdCo moments ("Methods"). Before we applied the current pulses, a large set field in the z-direction ($B_{set,z} = \pm 100$ mT) was applied to initialize the magnetic moments to a known state. After the moments are initialized, the large set field is removed. The current pulse amplitude sequence for studying the SOT switching behavior in Fig. 3 starts from a large negative current value (− 27 mA) above the critical switching threshold and traverses to the positive critical switching current value (+ 27 mA) and back to the negative critical switching current value with a hysteresis profile.

SOT switching curves from sample-45 without symmetry-breaking fields are shown in Fig. 3b. Switching between two magnetic states is observable, which confirms the field-free switching behavior in largely canted GdCo moments. In addition to the field-free switching, the $\Delta R_{AHE}$ appears to be dependent on the initialization direction with the case of $B_{set,z} = 100$ mT having a larger $\Delta R_{AHE}$, and the switching chirality is independent of the initialization direction. This indicates that the symmetry breaking caused by the canting is fixed in a particular direction, but the strength of the canting is dependent on the initialization direction. The difference in $\Delta R_{AHE}$ is amplified under a certain combination of $B_{set,z}$, and $B_x$. In Fig. 3c and d, the switching curves were obtained with ± 2.5 mT symmetry-breaking fields during the switching experiments, and the magnet was initialized with

$B_{set,z} = \pm 100$ mT. From these four switching curves, we can confirm that the effective canting field in $Gd_xCo_{100-x}$ is in the direction of $-\hat{x}$ since the switching chirality is the same between the case of $B_x = 0$ and the case of $B_x = -2.5$ mT. By comparing the four curves, the switching chirality is consistent with the symmetry-breaking field sign. However, for the case of $[B_{z,set}, B_x] = [100$ mT, 2.5 mT] (see Fig. 3c), the $\Delta R_{AHE}$ is highly suppressed compared to the rest of the three switching curves. While it is expected that a positive $B_x$ may partially cancel out the canting since the effective canting field is in the $-\hat{x}$ direction, it is not immediately clear on the $B_{z,set}$ dependence of $\Delta R_{AHE}$ suppression.

In order to better understand how the symmetry breaking field ($B_x$) and the initialization conditions affect the canting of the anisotropy during SOT switching, we repeated the switching experiment over a series of $B_x$ for both initialization cases $B_{set,z} = \pm 100$ mT (see Fig. 4). In Fig. 4, all $R_{AHE}$ loops are plotted as they are without any centering or normalization since the DC offsets of the $R_{AHE}$ are dependent on $B_x$. For moments initialized by 100 mT, clear switching due to SOT can be clearly seen in all symmetry-breaking fields (see Fig. 4a). However, reduced $\Delta R_{AHE}$ are present for $B_x = 1$ mT and 2.5 mT (See Figs. 4a and 3c), indicating partial switching. This can be due to the canting effective field being canceled by the externally applied in-plane field ($B_x$). The effective canting field from Fig. 4a is between 1 and 2.5 mT in the direction of $-\hat{x}$. This is further supported by the fact that the switching chirality changed between 1 mT and 2.5 mT. In a usual SOT switching framework without any canting in the anisotropy[21,22],

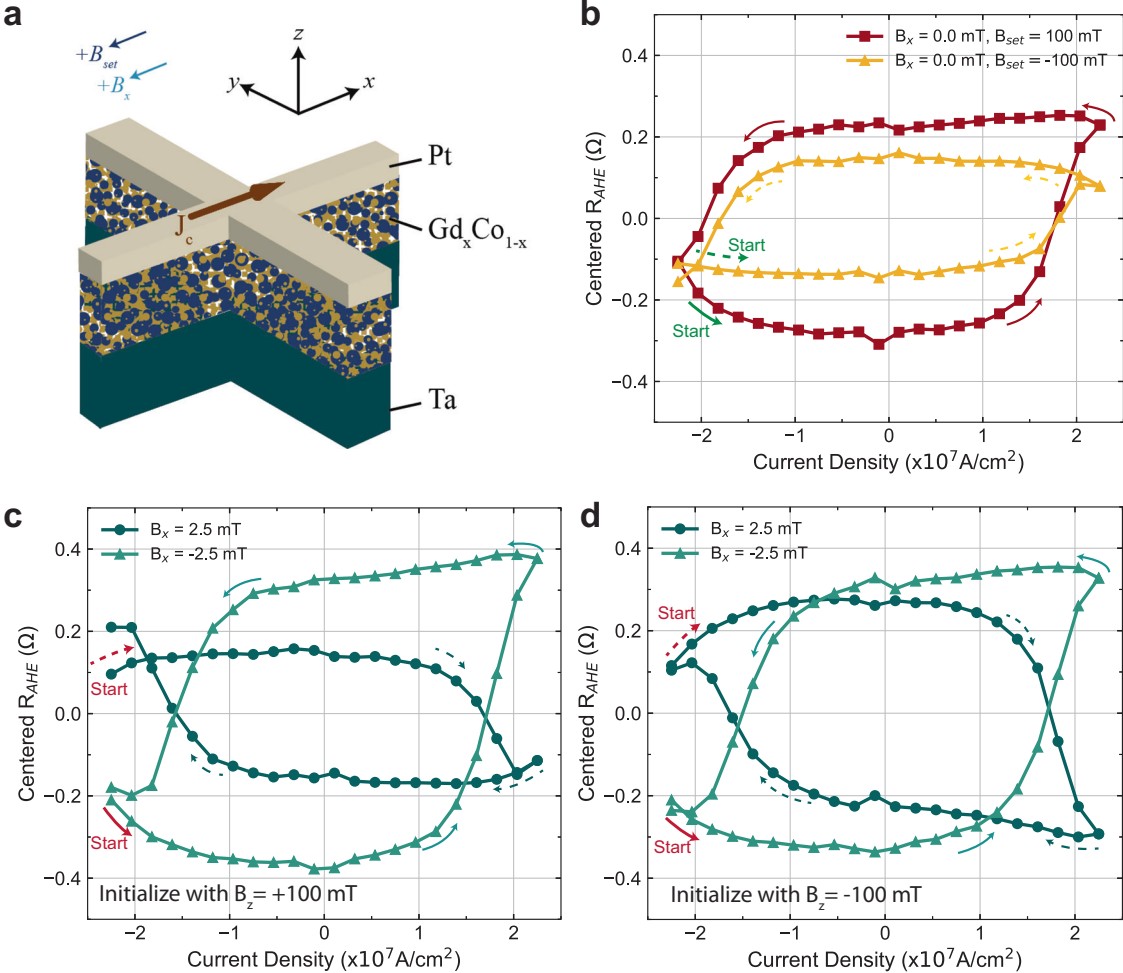

**Fig. 3 | Field-free spin-orbit torque switching of canted GdCo moments in sample with 45° canting angle (sample-45) with $B_x$ and initial magnetic state set by a large out-of-plane magnetic field ($B_z$). a** Schematics for switching experiment configuration on the Hall bar device. **b** Spin-orbit torque switching without a symmetry breaking field on sample-45 with anomalous Hall effect resistance ($R_{AHE}$) as the readout. Clear switching of the canted moments for both initialization conditions $B_{set,z} = 100$ mT and $-100$ mT is apparent with the case of $B_{set,z} = -100$ mT having a smaller $\Delta R_{AHE}$. The start of the current pulse is denoted with a green arrow in both cases (dashed for $B_{set,z} = -100$ mT). **c** Spin-orbit torque switching of canted GdCo moment in sample-45 with initialization field of 100 mT and $B_x = \pm 2.5$ mT. The start of the current pulse is denoted with a red arrow in both cases (dashed for $B_x = 2.5$ mT). **d** Spin-orbit torque switching of canted GdCo moment in sample-45 with initialization field of $-100$ mT and $B_x = \pm 2.5$ mT. The start of the current pulse is denoted with a red arrow in both cases (dashed for $B_x = 2.5$ mT).

switching chirality reverses when the sign of the symmetry-breaking field reverses. However, in Fig. 4a, the switching chirality did not reverse upon crossing the zero symmetry breaking field, instead, the chirality reversed between 1 and 2.5 mT. As for the moments initialized by $-100$ mT (see Figs. 4b and 3d), similar switching behavior is observed in the case of $B_{set,z} = 100$ mT. However, the field where the switching chirality reverses is reduced to be between 0 and 1 mT, with the $\Delta R_{AHE}$ to be heavily diminished at $B_x = 1$ mT.

From the $B_x$ dependent switching experiment (see Fig. 4), we found the effective symmetry-breaking field due to canting is between 0 and 2.5 mT in the $-\hat{x}$ direction, and it is also dependent on the initialization direction ($\pm \hat{z}$) with $B_{set,z} = 100$ mT yielding a larger canting effective field. This can be self-consistently observed in the amplitude of the $\Delta R_{AHE}$ at zero symmetry breaking field. Since the canting is weaker when the magnetic state is set with a $-B_z$ compared to $+B_z$, this means it is more difficult to achieve deterministic switching in the cases of $-B_z$. As a result, the $\Delta R_{AHE}$ at $B_x = 0$ will be smaller for the case of $B_{set,z} = -100$ mT. This also explains the diminished $\Delta R_{AHE}$ in Fig. 3c for the case of [$B_x = 2.5$ mT, $B_{set,z} = 100$ mT]. Another observation from this experiment is the flatness of the up and down magnetic states in the switching curves as a function of the

symmetry-breaking field. Regardless of the initialization field direction, the magnetic state that is set by a positive current in both switching chirality is always sloped in the current sweep direction that goes from positive to negative. This is visible in the upstate for $B_x \leq -2.5$ mT and the downstate for $B_x \geq 2.5$ mT in Figs. 4a, 3c. This has been observed in the AFM / FM bilayer system with an in-plane exchange bias field acting on the FM with PMA in past studies[23]. The fact that this behavior only shows up in one magnetic state rather than both can be due to the exchange spring effect and the ratchet effect in SOT, which is an asymmetry in the magnetic state stability[35]. In addition to the observations in Fig. 4, we repeated the $B_x$ dependent switching experiment with 27 mA instead of $-27$ mA in both initialization cases $\pm 100$ mT (Supplementary Fig. S2) and found even more distinct features. By starting the current pulse sequence from 27 mA, we found that for $B_{set,z} = 100$ mT case, a gap in $R_{AHE}$ exists when the SOT tries to switch the canted moments back to the initial state for $Bx \geq 5$ mT. Similar behavior can be observed in the case of $B_{set,z} = -100$ mT when $B_x \leq -2.5$ mT while the rest of the switching curves resemble the case in Fig. 4 correspondingly. To try to understand such a gap in $R_{AHE}$, we carried out more in-depth switching experiments.

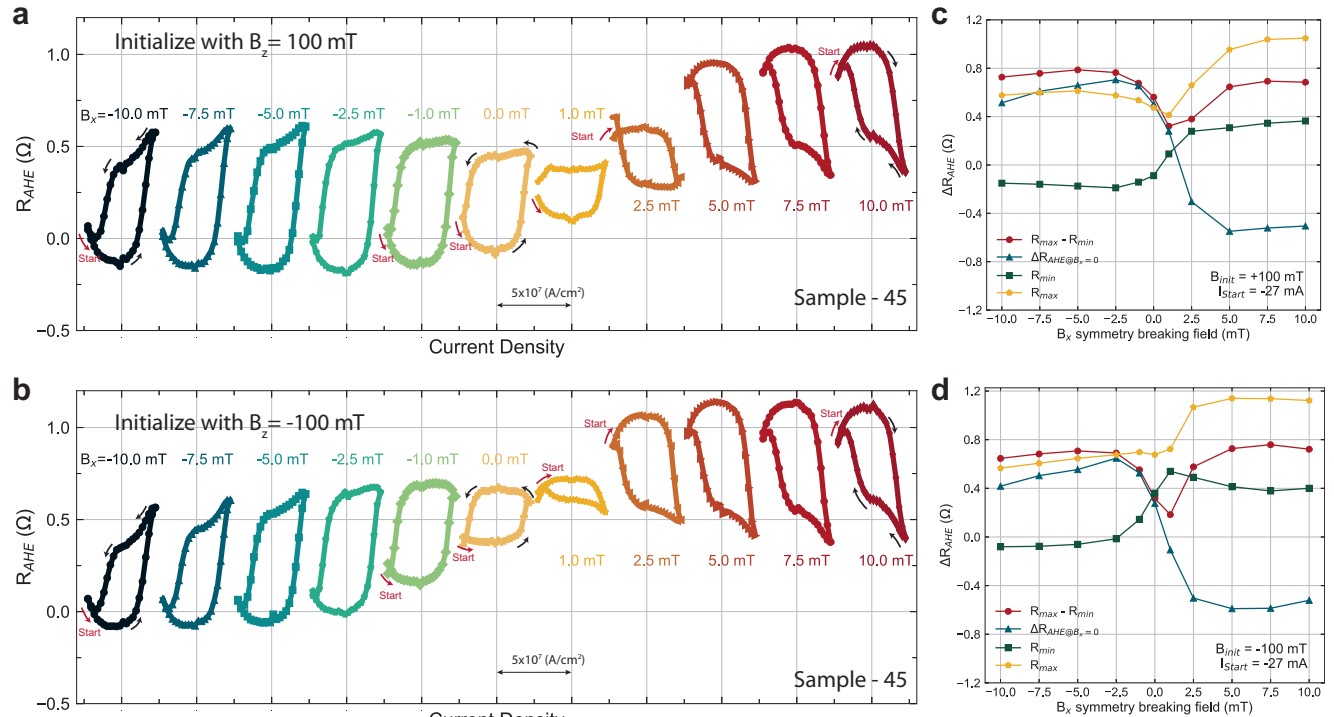

**Fig. 4 | In-plane symmetry breaking field ($B_x$) dependence on SOT switching of GdCo moments with 45° anisotropy canting angle (sample-45). a, b** SOT switching curves with varying symmetry breaking field ($\mathbf{B}_x$) from −10 mT to 10 mT. Before each switching experiment, the initialization field ($\mathbf{B}_{set,z}$) 100 mT and −100 mT were applied to initialize the moments into a known state, respectively.

**c, d** Extracted information of each switching curve as a function of symmetry breaking field for the case of $\mathbf{B}_{set,z} = 100$ mT and −100 mT, respectively. We extracted the $\Delta R_{AHE}$ of maximum and minimum $R_{AHE}$ throughout the entire switching curve, the $\Delta R_{AHE}$ of $R_{AHE}$ at the zero current crossing, the minimum $R_{AHE}$, and the maximum $R_{AHE}$.

## Tunable multi-state and ratchet effect in canted GdCo hetero-structures via spin-orbit torque

The ratchet effect in SOT is manifested in the asymmetry of SOT switching via a variety of transport signatures. It is mentioned in the previous section that only one magnetic state is sloped which is always set by the positive current in Fig. 4a, b. Another signature of the SOT ratchet effect is the sharpness of the switching between the two magnetic states. This is usually more prominent in systems with higher saturation magnetization, such as IrMn/CoFeB systems[35], nevertheless, it is still observable here in $Gd_xCo_{100-x}$. In Fig. 4a and b, for switching curves with decent switching amplitude ($|\mathbf{B}_x| > 1$ mT), the amount of $\Delta R_{AHE}$ per unit current density increase is larger when switching with positive current regardless of the chirality (down to up state for $-\mathbf{B}_x$ and up to the down state for $+\mathbf{B}_x$). In canted $Gd_xCo_{100-x}$, the ratchet effect in SOT can be more complex, where the exchange spring effect shows up in both in-plane and out-of-plane magnetic components. Since the initialization field direction affects the SOT switching behavior, including the effective canting field amplitude and the difference in $\Delta R_{AHE}$ from the $\mathbf{B}_x$ dependent switching experiment, it is important to understand what the magnetic state ($R_{AHE}$) has been set to by the initialization field ($\mathbf{B}_{set,z}$) before applying a large current pulse to switch the magnetic moments. As a result, we performed the $\mathbf{B}_x$ dependent switching experiment with a starting current that is small, so the SOT is negligible, and the initial magnetic state can be probed. Interestingly, the large initial set field $\mathbf{B}_{set,z}$ sets the magnetic moments into a state that is not reachable by SOT (see Fig. 5), and this initial state is successfully initialized to the same state regardless of the symmetry-breaking field applied during the switching experiment. This consistent initial state is quantitatively characterized by the constant $R_{AHE}$ value ($R_{min}$ for $\mathbf{B}_{set,z} = 100$ mT and $R_{max}$ for $\mathbf{B}_{set,z} = -100$ mT) in Fig. 5c, d. Thus, an additional state is visible, and this irreversible behavior manifests a similar SOT ratchet

effect in a single-layer RE-TM ferrimagnetic alloy as in AFM/FM systems[35,37,41].

The multistate and SOT ratchet effect can be observed in certain combinations of $\mathbf{B}_{set,z}$, and $\mathbf{B}_x$ in our study. In Fig. 5a, the initial magnetic states are consistently set to the lowest $R_{AHE}$ value around −0.25 Ω by $\mathbf{B}_{set,z} = 100$ mT across all the different $\mathbf{B}_x$-dependent scans, and in Fig. 5b, the initial magnetic states are set to the highest $R_{AHE}$ value consistently as well. Here, there are two observations we would like to point out, and they are differentiated by the switching chirality. In the first observation, we focus on the switching curves set by $\mathbf{B}_x \geq 2.5$ mT with $\mathbf{B}_{set,z} = 100$ mT, and $\mathbf{B}_x \leq 0$ mT with $\mathbf{B}_{set,z} = -100$ mT (Fig. 5a, b). In these cases, the current pulse amplitude increases to the first critical current value where SOT is strong, and the magnetic state is switched to the opposite direction away from the initial state. The reverse switching occurred upon reaching the second critical current with the opposite sign. However, the canted moments did not switch back to the original $R_{AHE}$ state set by the large initial field for cases of $\mathbf{B}_x \geq 2.5$ mT in Fig. 5a and $\mathbf{B}_x \leq 0$ mT in Fig. 5b. Instead, the $R_{AHE}$ state set by the SOT is near the mid-point of the full $R_{AHE}$ curve, exhibiting a ratchet behavior.

More interestingly, this ratchet effect exists for both switching chiralities. For the first observation, ratchet switching is defined in the fashion that the first critical current switched the $R_{AHE}$ state away from the initial state to the opposite state, and when the current amplitude reaches the second critical current with the opposite sign, the $R_{AHE}$ state that is set by the SOT does not return to the initial $R_{AHE}$ state set by the large magnetic field. For the second observation, we focus on the switching curves of $\mathbf{B}_x = -2.5 \sim -1$ mT with $\mathbf{B}_{set,z} = 100$ mT, and $\mathbf{B}_x = 1 \sim 5$ mT with $\mathbf{B}_{set,z} = -100$ mT. In this case, we still observe the multistate even though the first critical current sign already favors the initial magnetization state direction. Upon reaching the first critical current, the $R_{AHE}$ state is set to a different value from the initial $R_{AHE}$

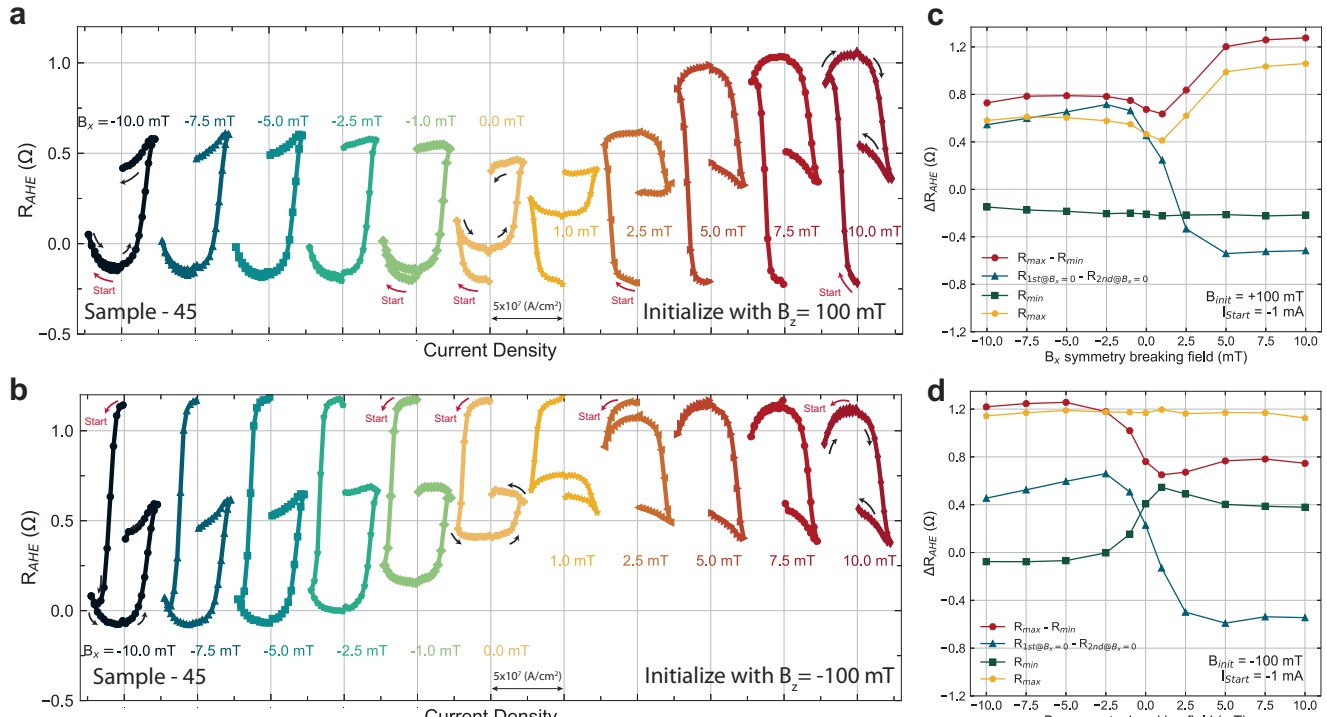

**Fig. 5 | Tunable multi-magnetic state switching and ratchet effect driven by spin-orbit torque as a function of in-plane symmetry breaking field ($B_x$) and initialization field ($B_{set,z}$). a, b** SOT switching curve with varying external symmetry breaking field ($B_x$) and initialization with $B_{set,z}$ of $+100$ mT and $-100$ mT, respectively. The current pulse sequence starts from a small negative current ($-1$ mA), then traverses in the negative direction to the negative critical switching current and back to the positive critical current density, then stops at a small positive current ($+1$ mA). **c, d** Extracted information of each switching curve as a function of symmetry breaking field for the case of $B_{set,z} = +100$ mT and $-100$ mT, respectively. We extracted the $\Delta R_{AHE}$ of maximum and minimum $R_{AHE}$ throughout the entire switching curve, the $\Delta R_{AHE}$ of $R_{AHE}$ at the zero current crossing, the minimum $R_{AHE}$, and the maximum $R_{AHE}$.

and once the current reaches the second critical current with the opposite sign, the $R_{AHE}$ state is switched to the opposite magnetic state. In fact, as the symmetry-breaking field becomes stronger, the middle $R_{AHE}$ state that is set by the first critical current shifts into the $R_{AHE}$ state that is set by the second critical current value, and the multistate is absent. This shows that the middle $R_{AHE}$ state can be tuned by the external symmetry-breaking field.

If we disregard the initial magnetic state in Fig. 5, the middle $R_{AHE}$ is essentially the bottom (top) state for the case of $B_{set,z} = 100$ mT ($-100$ mT) in Fig. 4. To facilitate the comparison, the superposition of the switching curves obtained with different starting current values but same symmetry breaking field and initialization field is plotted in Supplementary Fig. S3. The two states achieved by the SOT with a large starting current is exactly overlapping on the part where the middle state and the final state are achieved by the SOT in the curve with the multistate (see Supplementary Fig. S3). By calculating the $\Delta R_{AHE}$ for $[B_x, B_{set,z}] = [7.5$ mT, $100$ mT] and $[-7.5$ mT, $-100$ mT] in Fig. 5, we obtain $\Delta R_{AHE} = 1.26$ and $1.25$ $\Omega$ respectively (see Fig. 5c, and d). If we further compare these values to the $\Delta R_{AHE}$ at zero field ($1.064$ $\Omega$), obtained by multiplying the AHE field sweep hysteresis loop in the z-direction by $\sqrt{2}$ since it is detected by a lock-in amplifier (See Fig. 3b and methods), we find very similar values. This indicates that SOT either fully switched or very close to fully switched the canted moments from its initial state set by the large $B_{set,z}$ to the opposite state resulting in a $\Delta R_{AHE}$ close to the $\Delta R_{AHE}$ obtained by field sweep where the moments are fully saturated in the z-direction. However, only partial switching can be obtained when the moments are switched back, resulting in the presence of a middle state in Fig. 5, thus the ratchet effect. This phenomenon suggests the presence of fixed effective fields due to canting in both the in-plane and the out-of-plane direction. The change of the $R_{AHE}$ under different $[B_x, B_{set,z}]$ conditions

in Fig. 5 is the result of both the in-plane and the out-of-plane components of the canted GdCo moments are being manipulated by the SOT and the switching dynamics of both components affecting each other, potentially due to the exchange spring effect. For completeness and repeatability, the symmetry-breaking field dependence switching curves of conditions $[I_{start} = 1$ mA, $B_{set,z} = 100$ mT] and $[I_{start} = 1$ mA, $B_{set,z} = -100$ mT] are shown in Supplementary Fig. S4 and the results are consistent and symmetric to the cases in Fig. 5.

### Effect of different canting angles on spin-orbit torque switching in GdCo heterostructures

From the SOT switching results on sample-45, it is shown that strong magnetic anisotropy canting can lead to different switching phenomena, such as the ratchet effect, field-free switching, and multistate switching. Here we further explore the effect of canting angle on the various switching phenomena. From our sample preparations, single-shot co-sputtering deposition of GdCo can lead to different canting angles even with the same sputtering parameters but near the compensation point. Two other samples we found to possess different canting angles are shown in Fig. 6 (56.25°) and Fig. 7 (13.5°). Sample-56 has a much larger saturation magnetization (Fig. 6a), and the canting angle (56.25°) is closer to in-plane (Fig. 6c). For sample-56, the two-phase switching in the magnetometry data is weaker but still observable, especially in the OOP direction (Fig. 6a). Although the anisotropy is closer to being in-plane thus the smaller squareness ($M_r / M_s$), the field-sweep AHE curve still exhibits a clear hysteresis (Fig. 6b). However, the SOT switching curve does not exhibit a strong two-state switching, instead, a very gradual switching curve as a function of current amplitude is observed (Fig. 6d and e). This is mostly due to the weak PMA of sample-56 since the readout of the magnetic state is through the anomalous Hall effect. Interestingly, among the two

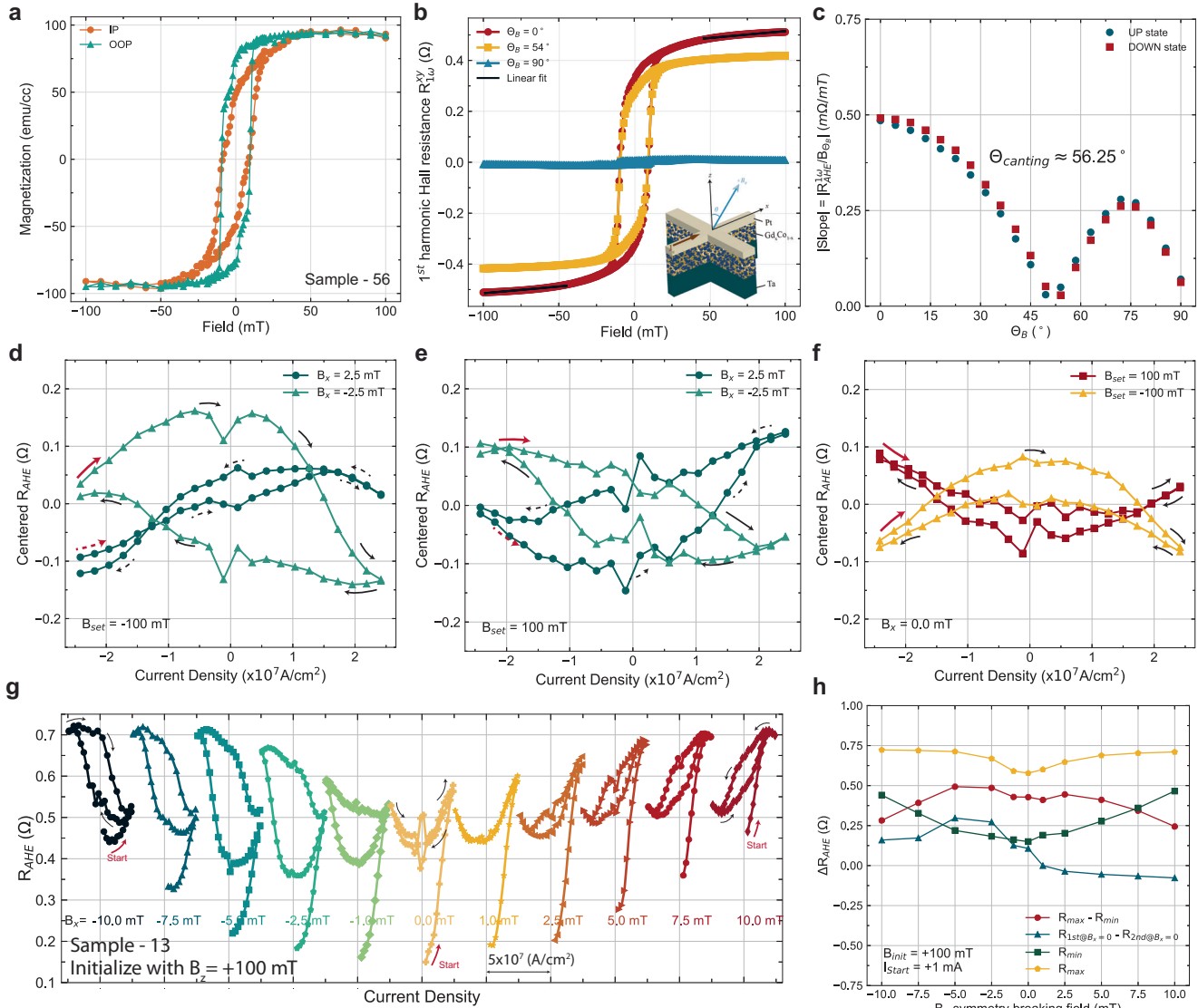

**Fig. 6 | SOT switching of canted GdCo moments with magnetic canting angle of 56 degrees (sample-56). a** Thin film magnetometry (Magnetization - applied magnetic field) of sample-56 via vibrating sample magnetometer measured in both in-plane and out-of-plane direction. **b** Angle-dependent anomalous Hall effect measurement and the linear fit to data above the coercive field region for obtaining the slope. **c** Slope in the field region above the coercive field as a function of field angle. The minimum slope resides at 56.25° from the $\hat{z}$-axis. **d, e** Spin-orbit torque switching curve with starting current value of − 27 mA. The four different switching curves are obtained with four different combinations of initialization field (±100 mT) and symmetry breaking field (±2.5 mT). **f** Spin-orbit torque switching curve with a starting current value of − 27 mA without a symmetry-breaking field. The two curves show the dependence of the initialization direction. **g** SOT switching curves with varying external symmetry breaking field ($\mathbf{B}_x$) and initialization with $\mathbf{B}_{set,z}$ of 100 mT. The current pulse sequence starts from a small positive current (1 mA), then traverses in the positive direction to the positive critical switching current and back to the negative critical current density, then stops at the positive critical current. The Ratchet effect is visible with the effect amplified at zero symmetry breaking field. The symmetry-breaking field-dependent SOT switching curve and ratchet effect of the other combinations [$I_{start}$ = ±1 mA, $\mathbf{B}_{set,z}$ = ±100 mT] are shown in Fig S5. **h** Extracted information of each switching curve as a function of symmetry breaking field for the case of $\mathbf{B}_{set,z}$ = 100 mT. We extracted the $\Delta R_{AHE}$ of maximum and minimum $R_{AHE}$ throughout the entire switching curve, the $\Delta R_{AHE}$ of $R_{AHE}$ at the zero current crossing, the minimum $R_{AHE}$, and the maximum $R_{AHE}$.

different switching conditions [$\mathbf{B}_{z,set}$ = −100 mT, $\mathbf{B}_x$ = ±2.5 mT], the $\Delta R_{AHE}$ of the case [$\mathbf{B}_x$ = +2.5 mT] is highly suppressed but largely enhanced for the case [$\mathbf{B}_x$ = −2.5 mT] (Fig. 6d). This is similar to sample-45 (Fig. 3c) where $\Delta R_{AHE}$ is suppressed for the case of [$\mathbf{B}_{z,set}$ = 100 mT, $\mathbf{B}_x$ = 2.5 mT] but the discrepancy between the opposite sign of $\mathbf{B}_x$ is enhanced with a larger canting angle (sample-56). For the cases of [$\mathbf{B}_{z,set}$ = 100 mT, $\mathbf{B}_x$ = ±2.5 mT] in sample-56, the $\Delta R_{AHE}$ is comparable regardless of the symmetry breaking field, and also the chirality switches accordingly, which is the same as the case [$\mathbf{B}_{z,set}$ = −100 mT, $\mathbf{B}_x$ = ±2.5 mT] (Fig. 3d) in sample-45.

While it is expected that very minimal magnetic switching can be observed in sample-56 without a symmetry-breaking field due to the weak PMA, a non-zero hysteresis is still observed at zero symmetry-breaking field (Fig. 6f). In addition, the switching chirality is fixed regardless of the initialization field (Fig. 6f), which is also similar to the zero-field switching result in sample-45 (Fig. 3b). The opposite sign of the curvature in the two switching curves (Fig. 6f) is most likely due to the IP moments that are initialized by the opposite perpendicular field. Here, we also observed a minimal hysteresis opening at $\mathbf{B}_x$ = 1 mT, which indicates that the canting effective field is close to 1 mT and also the switching chirality changes upon crossing $\mathbf{B}_x$ = 1 mT. Next, we investigate how the canting angle affects the multi-state switching and ratchet effect. Clearly, the ratchet effect persists in all the switching curves with different $\mathbf{B}_x$ (Fig. 6g), and the main

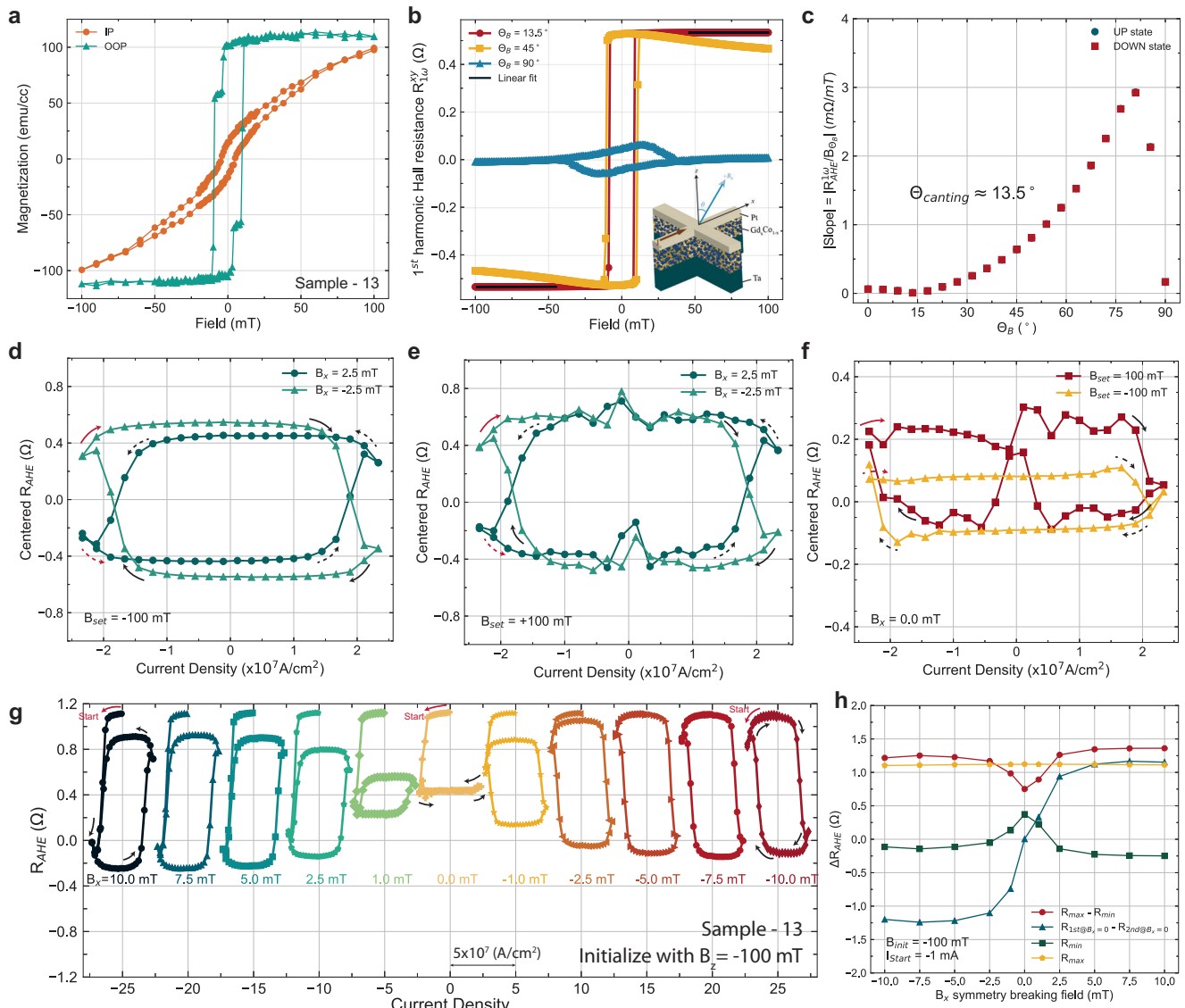

**Fig. 7 | SOT switching of canted GdCo moments with magnetic canting angle of 13 degrees (sample-13). a** Thin film magnetometry (Magnetization-applied magnetic field) of sample-13 via a vibrating sample magnetometer measured in both in-plane and out-of-plane direction. **b** Angle-dependent anomalous Hall effect measurement and the linear fit to data above the coercive field region for obtaining the slope. **c** Slope in the field region above the coercive field as a function of field angle obtained from the angle-dependent anomalous Hall measurement. The minimum of the slope resides at 13.5° from the z-axis. **d, e** Spin-orbit torque switching curve with starting current value of − 28 mA. The four different switching curves are obtained with four different combinations of initialization field (±100 mT) and symmetry breaking field (± 2.5 mT). **f** Spin-orbit torque switching curve with a starting current value of − 27 mA without a symmetry-breaking field. The two curves show the dependence of the initialization direction. **g** SOT switching curves

with varying external symmetry breaking field ($\mathbf{B}_x$) and initialization with $\mathbf{B}_{set,z}$ of − 100 mT. The current pulse sequence starts from a small negative current (− 1 mA), then traverses in the negative direction to the negative critical switching current and back to the positive critical current density, then stops at the negative critical current. The Ratchet effect is visible with the effect amplified at zero symmetry breaking field. The symmetry-breaking field-dependent SOT switching curve and ratchet effect of the other combinations [$I_{start} = ± 1$ mA, $\mathbf{B}_{set,z} = ± 100$ mT] are shown in Supplementary Fig S6. **h** Extracted information of each switching curve as a function of symmetry breaking field for the case of $\mathbf{B}_{set,z} = 100$ mT. We extracted the $\Delta R_{AHE}$ of maximum and minimum $R_{AHE}$ throughout the entire switching curve, the $\Delta R_{AHE}$ of $R_{AHE}$ at the zero current crossing, the minimum $R_{AHE}$, and the maximum $R_{AHE}$.

difference from sample-45 is that the initial magnetic state ($R_{AHE}$) is no longer constant (Fig. 5) across the different symmetry-breaking fields in sample-56. In fact, the initial state $R_{min}$ (Fig. 6h) is symmetric against $\mathbf{B}_x = 0$. As for the ratchet effect, it is clear that once the SOT switches the moments to the other direction, the opposite SOT sign does not switch the moments back to their initial state $R_{min}$ (Fig. 6g). Full $\mathbf{B}_x$-dependent switching curves with different current loop traverse directions and initialization directions are shown in Supplementary Fig. S5. Due to the weak PMA in sample-56, the effective canting field is smaller (between 0 ~ 1 mT) than sample-45 across all the sets ([$\mathbf{B}_{z,set}$, $\mathbf{B}_x$]) of switching curves.

Next, we investigate the effect of a much weaker canting on the switching phenomena. A strong PMA can be obtained with clear exchange spring behavior from the magnetometry measurement in the OOP direction for single-shot co-sputtered GdCo with its composition close to the magnetic compensation - sample-13 (Fig. 7a). From the angle-dependent AHE field sweep measurement (Fig. 7b, c), the canting angle is close to 13°. The discrepancy of $\Delta R_{AHE}$ between the two $\mathbf{B}_x = ± 2.5$ mT switching curves, when the moments are initialized by $\mathbf{B}_{z,set} = − 100$ mT, is observable but much smaller (Fig. 7d) compared to sample-45 (Fig. 3b) and sample-56 (Fig. 6d). As for the other initialization case ($\mathbf{B}_{z,set} = 100$ mT), the $\Delta R_{AHE}$ are the same between the two

$B_x = \pm 2.5$ mT switching curves and their chiralities behave as expected correspondingly (Fig. 7e). Due to the weaker canting and strong PMA, minimal switching is observed at zero symmetry breaking field (Fig. 7f) and the switching chirality is again fixed. Although the canting is weak, a clear SOT ratchet effect is observable across the different symmetry-breaking field switching loops (Fig. 7g). However, due to the strong PMA and weak canting, the switching curves and multistate behavior are less tunable across the different in-plane symmetry breaking fields. For example, in sample-45, the switching curves as a function of in-plane field ($B_x$) resided in the $R_{AHE}$ range of 0 to 0.5 $\Omega$ for $B_x = -10$ mT, and as $B_x$ increases, the switching curve moved up in $R_{AHE}$ values and end up in the range of 0.5 to 1.2 $\Omega$ for $B_x = 10$ mT. This can be characterized by the $R_{min}$ (Fig. 4d) and $R_{max}$ (Fig. 4d) in Fig. 4b. This is not the case for sample-13 where the switching curves do not move up by more than 0.125 $\Omega$ from $B_x = -10$ mT to 10 mT in Fig. 7g and characterized by $R_{min}$ in Fig. 6h. For completeness, $B_x$-dependent switching curves of different $B_{z,set}$ and different current start values are shown in Supplementary Fig. S6. A clear ratchet effect can be observed in all of the switching curves.

Since the IP magnetization is crucial to observations such as field-free switching, multistate and ratchet effect, we performed a simple $B_y$ IP field dependent SOT switching experiment to determine whether a significant portion of the IP magnetization lies along the $y$-direction, which is transverse to the current direction. According to Kong et al.[42], if the IP magnetization lies in the direction transverse to the current direction, strong SOT switching can be observed regardless of the applied transverse ($B_y$) IP field. However, this is not the case for our system as minimal albeit non-zero SOT switching can be observed as $B_y$ increases (Supplementary Fig. S8). This indicates that the majority of the IP magnetization lies in the direction of the current ($x$) with a small tilt toward the y-direction. We also studied the possibility of the memristor behavior[23,34] in sample-13 since the strong PMA can lead to much easily observable SOT switching loops and $R_{AHE}$ value changes. Due to the strong exchange spring effect in the OOP direction and the small canting, GdCo in sample-13 serves as an equivalent system as AFM/FM, which possesses a small canting from the exchange bias and the exchange spring behavior from the coupling between the pinned interface AFM moments and FM moments. Indeed, the memristor behavior can be observed in sample-13 with a small symmetry-breaking field to assist full switching (Supplementary Fig. S7). In addition, the spacing between the $R_{AHE}$ states achieved by the different $I_{max}$ values can be tuned by the symmetry-breaking field strength.

## Discussion

Our results have demonstrated that unconventional SOT switching behaviors, including multistate, memristor, field-free, and ratchet SOT switching, can be achieved by engineering appropriate anisotropy-induced canting in ferrimagnetic Gd$_x$Co$_{100-x}$. In addition to the canting, the presence of both a hard and a soft magnetic phase in each of the magnetic anisotropy directions (IP and OOP) exhibits characteristics of an exchange-spring system. At the same time, the ratchet effect in SOT switching provides further confirmation of an exchange-spring system. The tunability of such multi-state and ratchet effects is highly dependent on the canting of the magnetic anisotropy. A canting angle of ~ 45° is the ideal system to have tunable multistate ($R_{AHE}$) and large $\Delta R_{AHE}$, as it was demonstrated in this work. The ratchet SOT effect is observed across all canting angles, and it exists regardless of the symmetry-breaking field or initialization direction. These diverse SOT switching effects can be useful for applications such as neuromorphic computing[43,44], multistate magnetic memory, and efficient magnetic memory.

## Methods
### Sample preparation and device fabrication
Thin film Ta/GdCo/Pt heterostructure is deposited by magnetron sputtering at room temperature. Specifically, the GdCo layer is deposited by co-sputtering Gd and Co elemental targets in one single shot with fixed calibrated powers. Co power is fixed at 60 W, while Gd power is varied between 20 and 30 W for different concentrations. No pause or change of the Gd and Co sputtering powers during the growth is implemented. The heterostructure is deposited on thermally oxidized silicon of 100 nm formed on a silicon substrate. All three samples have the stack design as Si substrate / SiO2 (100 nm) / Ta (8 nm) / Gd$_x$Co$_{100-x}$ (10 nm) / Pt (2 nm). The concentration of each Gd$_x$Co$_{100-x}$ sample is shown in Table 1. Hall bar devices are patterned with standard photolithography and ion milling. Metal contacts to the device are fabricated through the lift-off process with electron beam evaporation of Ti (5 nm) / Au (80 nm).

### Oxygen plasma treatment and diffusion for inducing magnetic canting
Oxygen plasma treatment is carried out with parameters including 50 W RF power, 180 mTorr oxygen, and 50 °C chuck in a Technics C Plasma Etching System in the Berkeley Nanofabrication facility. Diffusion of oxygen is done under ambient conditions of room temperature and atmospheric pressure. More explanation can be found in Supplementary Note 2.

### Magnetometry and anomalous Hall measurements
Thin film magnetometry measurements are done at room temperature with a Lakeshore 7400 series vibrating sample magnetometer in both in-plane and out-of-plane directions. Analysis of the magnetometry result is done by subtracting out the diamagnetic signal from the silicon substrate and shifting the DC offset of the curve to zero moments. The diamagnetic signal from the silicon is obtained by fitting a line in the saturated regime. Magnetization is calculated by dividing the measured moments by the volume of the thin film GdCo.

Angle-dependent anomalous Hall field sweep measurements are done under room temperature with a home-built setup that consists of an Amtek 7270 lock-in amplifier, a Keithey 6221 ac current source, a Lakeshore 475 Gaussmeter, and a GMW 5403 electromagnet driven by a kepco power supply that can produce a bi-polar magnetic field up to 330 mT. Magnetic fields are applied with a hysteresis profile, and the AC current is fixed at a small value (500 μA), so minimal spin-orbit torque affects the magnetization. The current direction is fixed at the $x$-direction and the magnetic field direction starts from the $z$-direction as 0° and traverses to the $x$-direction as 90°, which is the in-plane direction. Analysis of the angle-dependent anomalous Hall field sweep measurement involves the fitting a linear function to the data in the region above the coercive field for both up and down magnetic states. By extracting the slope of such linear function, we can compare the angle of the easy-axis in our Gd$_x$Co$_{100-x}$ magnetic devices.

### Pulse-IV switching measurements
Pulsed I−V switching measurements are carried out with a home-built setup including a Keithley 6221 ac current source, a Keithley 2182a nanovoltmeter, a GMW 5403 electromagnet, a Lakeshore 475 Gaussmeter, and a custom-built rotating dip stick chip carrier with electrical contacts. Keithley 6221 and 2182a are connected together in the pulsed-delta mode for simultaneous current sourcing and voltage measurement. All the switching measurements are conducted with 200 μs pulse width and 60 μs source delay. Every pulse switching curve is obtained with a reset magnetic field applied in the $\pm \hat{z}$-direction at 100 mT before the current pulse sequence is applied. After the large set field is applied, the field is removed, and the sample is rotated back to in-plane configuration with respect to the magnetic field application direction ($\hat{x}$-direction). The in-plane symmetry breaking field is set and kept on throughout the entire pulsed I−V sequence. Voltage is measured in the transverse direction utilizing the anomalous Hall effect to read out the magnetic state. The SOT pulsed I−V measurements are done on Hall bar devices with 6 μm width and 40 μm

length. The Hall voltage arms of the Hall bar devices are 2.5 μm in width and 14 μm in length.

## Data availability

The authors declare that all data supporting the claims of this work including every data point collected are included in the main text, methods and the supplementary information. All raw data in table format are available at https://doi.org/10.6084/m9.figshare.27020875. Additional physical materials can be requested from S.S. (sayeef@berkeley.edu) or C.-H.H. (chhsu@berkeley.edu).

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

## Acknowledgements

This work was primarily supported by the U.S. Department of Energy, Office of Science, Office of Basic Energy Sciences, Materials Sciences and Engineering Division, under Contract No. DE-AC02-05-CH11231 within the Non-Equilibrium Magnetism program (MSMAG). This work was also supported in part by the ASCENT center, one of the six centers within the JUMP initiative jointly supported by DARPA and SRC. In addition, support from NSF E3S center is gratefully acknowledged. This work was performed in part at the Berkeley Marvell Nanofabrication Laboratory at the University of California, Berkeley, and their staff research support is greatly appreciated.

## Author contributions

The original idea was conceived by C.-H.H. and the research project is supervised by S.Salahuddin. Film preparation was performed by C.-H.H. and M.J.G.; device fabrication was performed by C.-H.H.; magnetic characterization was performed by C.-H.H. and H.C.K.; transport and switching measurement was performed by C.-H.H.; data analysis was done by C-H.H. under the supervision of S.Salahuddin; C.-H.H. and S.Salahuddin co-wrote the manuscript. All authors contributed to discussions and commented on the manuscript.

## Competing interests

The authors declare no competing interests.
