## [Peer Review File · Nature Communications]

Reviewers' Comments:

Reviewer #1:

Remarks to the Author:

This manuscript reported field-free switching, multistate switching, memristor behavior and ratchet effect in $[(\text{Gd})_x(\text{Co})_{100-x}]$ structures. A large anisotropy canting can be observed in $[(\text{Gd})_x(\text{Co})_{100-x}]$ systems. A large sum of testing data has been summarized and analyzed, including many important aspects such as multistate switching, memristor behavior and ratchet effect. These results may be helpful for people working in the field of spintronics. However, two concerns should be addressed before I can suggest its publication:

1. In structures with canting anisotropy, field-free SOT switching can be realized in the absence of a symmetry breaking field. The symmetry breaking is induced by the canting anisotropy. It's easy to understand the field-free SOT switching in sample-45. But for sample-56, field-free SOT switching is not applicable (Fig.5f). That's to say, to achieve field-free switching, a proper canting angle is necessary, although the canting angle difference between sample-56 and sample-45 is not large. But it's puzzling that sample-13, with a much stronger perpendicular magnetic anisotropy (PMA) and a weaker symmetry breaking than sample-56, can realize field-free SOT switching. As shown in Fig.6a, a strong PMA can be observed in sample-13, but 13° canting is still enough to induce field-free switching. It will be more convincing if more devices with different canting angles can be supplemented.

2. As mentioned above, the anisotropy canting angle of $[(\text{Gd})_x(\text{Co})_{100-x}]$ structures is crucial for field-free SOT switching and PMA. If the canting angle could be tuned easily, we can find the best canting angle to obtain proper PMA, field-free switching and other functions like multistate switching and ratchet effect. Therefore, the key is if the author could tune the anisotropy canting angle and obtain specific canting angle. If the canting angle can be tuned, the methods and results need to be stated more clearly.

Reviewer #2:

Remarks to the Author:

Dear Editor:

This work investigated the SOT effect of ferrimagnetic GdCo alloys from the perspectives of field-free, anisotropy, and memristivity. The observed phenomenon is novel but arguable. I believe the manuscript remains serious problems as follows.

1. I believe the most controversial point of this manuscript has to do with alloy composition. In Table I, how come sample-45 and -56 with identical composition exhibit

dramatically different properties? Also, how come a minor change of the composition from Co_{75.3} (sample-45 and -56) to Co₇₅ (sample-13) would result in a huge difference in magnetic properties? The readers feel confused while linking Fig. 4, 5, and 6 with Table I. The authors claimed that this is due to the compensation point where magnetic competition occurred within the bulk ferromagnet, but my argument is whether the observed goodness (field-free, memristivity, etc) is highly uncertain and uncontrollable? Note that any kind of magnetic competition, whether generated by interface exchange interaction or bulk sublattice, would create some degree of magnetic canting and instability, which could be the source of the phenomenon observed by the authors; i.e., the applied current breaks the symmetry of such kind of magnetic competition? Besides, even the Joule heating effect could disturb the spin competition and temporarily reverse the magnetic dominance from Gd and Co. In other words, the goodness generated by such conditions is in fact very unstable and difficult to manipulate.

2. I suggest the authors provide a schematic illustration that summarizes the finding. The relationship between initial magnetization, applied field, and SOT write pulse, etc is (Fig2,3,4) not reader-friendly and hard to conclude from a quick glance.

3. Is it possible to retrieve information about the in-plane magnetization component during SOT switching? This considers that the in-plane magnetization component plays a crucial role in ratchet behaviors.

4. Following #3, as the magnetic easy axis is tilted away from the z-axis, its in-plane component is critical to SOT switching. If it is strictly parallel to SOT write current, then it can be considered an effective H_x . However, if there is non-negligible in-plane magnetization component that is perpendicular to SOT write current, the situation can be much more complex. It reminds me of the type-T tri-layer structure reported by W. J. Kong et al., Nat Commun 10, 233 (2019); both cases yield field-free switching, but the underlying mechanisms seem to be different. Therefore, it is important to verify the orientation of the IP component of the canted moment. Although the authors did not mention it in the main text, it seems that they considered that the IP magnetization component is parallel to SOT write current. Also, it would be interesting to investigate the in-plane angular dependence of SOT switching by varying the relative orientation of electrical current with respect to the in-plane magnetization component.

5. In the studied structure (substrate/Ta/GdCo/Pt), I see three possible sources that generate SOT: magnetization-independent SHE from Ta and Pt, magnetization-independent SHE from GdCo, and magnetization-dependent SHE from GdCo (or anomalous spin Hall effect). There might be certain SOT components that are actually dependent on magnetization, quantitative estimation of effective torque under different

magnetization orientations will give more insight. Some observed phenomena may be the combined effect of magnetization-dependent SOT, tilted anisotropy, and exchange spring effects. The authors should clarify them.

6. I don't see supplementary note 1 where the authors are supposed to explain the Gd symmetric 4f filling. This is important as for a symmetrically filled 4f orbital there should exist no magnetic anisotropy, but this is not the case in this work. Please explain.

Reviewer #3:

Remarks to the Author:

The authors have presented a detailed study of CoGd ferrimagnetic system, emphasizing on three main behaviour - field free switching, ratchet behaviour and multistate switching. The authors have done an excellent job in convincingly demonstrating the above three behaviour. However, due to lack of novelty in terms of the SOT system presented and in terms of observed behaviour, I cannot recommend the manuscript for publication in Nature Communications. Field free switching has been observed and reported in ferrimagnetic system earlier (Zheng et al. Nature comm). In fact this field free switching has been observed in single layer ferrimagnetic system without any SOC layer. Similarly, multi level switching been observed in many ferromagnetic system. A similar observation in ferrimagnetic system seems to be an obvious extension and does not entail much novelty. Beside the above, I would request the author to include the recipe or work towards engineering the canted state. At present it seems like that random canted states are obtained during deposition.

Response to referee comments

for manuscript NCOMMS-22-52688

Firstly, we appreciate all three referees taking time to provide these invaluable suggestions and thoughtful comments. We took these suggestions and performed more experiments to strengthen the manuscript and address the referees' comments. We focused on addressing the three major referees' concerns which include the mechanism and method to induce canting in GdCo / heavy metal heterostructure, the role of in-plane magnetization in the canted magnetic heterostructure, and the repeatability. A summary of new experiments and data are provided followed by a list of modifications to the manuscript. Lastly, we provide the detailed point-by-point response to the Referees' comments.

Brief summary of modifications to the manuscript

0.1 Mechanism and Method of inducing canting

- **New Experiments:** Oxygen plasma treatment on the Ta/GdCo/Pt heterostructure for enhancing perpendicular magnetic anisotropy thus the magnetic canting. Confirmation of the canting was characterized by the anomalous Hall resistance field sweep measurement.
- **New/modified Figure:** New Figure 2 is added to the main text on inducing the canting via oxygen plasma treatment.
- **New/modified Text:** New paragraph on page 10
- **New/modified Methods section:** section "Oxygen plasma treatment and diffusion for inducing magnetic canting"
- **New/modified supplementary:** Supplementary Note 2

0.2 Role and direction of IP magnetization

- **New Experiments:** We conducted B_y -dependent (transverse IP field) SOT switching on 13.5° canting sample to study the IP magnetization direction during switching.
- **New/modified Figure:** New supplementary figure 8
- **New/modified Text:** New text in the last paragraph on page 27

Response to individual referee comments

All responses are in blue.

Referee 01

Comment

This manuscript reported field-free switching, multistate switching, memristor behavior and ratchet effect in $\text{Gd}_x\text{Co}_{100-x}$ structures. A large anisotropy canting can be observed in $\text{Gd}_x\text{Co}_{100-x}$ systems. A large sum of testing data has been summarized and analyzed, including many important aspects such as multistate switching, memristor behavior and ratchet effect. These results may be helpful for people working in the field of spintronics. However, two concerns should be addressed before I can suggest its publication:

Response

We appreciate that the referee found our results helpful to the spintronics community. We find the suggestions from the referee fruitful which greatly improve this study in both technical and communicative aspects. We have included follow-up experiments and explanations to address the concerns. Please see responses below.

Comment

1. In structures with canting anisotropy, field-free SOT switching can be realized in the absence of a symmetry breaking field. The symmetry breaking is induced by the canting anisotropy. It's easy to understand the field-free SOT switching in sample-45. But for sample-56, field-free SOT switching is not applicable (Fig.5f). That's to say, to achieve field-free switching, a proper canting angle is necessary, although the canting angle difference between sample-56 and sample-45 is not large. But it's puzzling that sample-13, with a much stronger perpendicular magnetic anisotropy (PMA) and a weaker symmetry breaking than sample-56, can realize field-free SOT switching. As shown in Fig.6a, a strong PMA can be observed in sample-13, but 13° canting is still enough to induce field-free switching. It will be more convincing if more devices with different canting angles can be supplemented.

Response

This is a very good point. We would like to clarify that all three samples exhibit some field-free switching albeit the opening of the R_{AHE} loop has different magnitude. Even for sample-56, a small hysteresis opening can be observed in Fig. 6f and the opening is larger in the case of 0 mT as compared to 1 mT in Fig. 6g. However, we would like to clarify a potential miscommunication. The canting angle mentioned in the manuscript is referring to the easy-axis tilting away from the out-of-plane in the net magnetization of the whole magnetic device fabricated from the GdCo heterostructure instead of the canting angle of the out-of-plane magnetization. This is the case due to the angle dependent AHE field sweep technique we employed to characterize the canting angle. This technique locates the canting angle at the minimal slope in the field region before saturating the entire magnetization of the AHE curve. This easy-axis canting of the net magnetization does not equal to the tilting of the OOP magnetization component in the GdCo layer. It depends on a few more factors including the magnetization of the IP and OOP component respectively, and the anisotropy strength of each component.

To elaborate in more detail, the slope of the AHE curve in the region before saturation can be contributed by two factors: 1. tilting of the OOP magnetization component (M_{oop}) away from the OOP axis and 2. tilting of the IP magnetization component (M_{ip}) away from the IP axis (Fig. R1b). The first factor reduces the AHE signal since AHE is proportional to the amount of OOP magnetization whereas the second factor will increase the AHE signal as more IP moments are pulled toward the

Figure R1: (a) Schematic of the magnetization components including the out-of-plane (OOP) and in-plane (IP) components in the GdCo layer. (b) Tilting of the IP and OOP magnetization components toward the external field direction. The external field is applied at an angle close to 45° from the OOP direction.

OOP direction. Depending on both the magnetization and the anisotropy strength of each component, a magnetic field applied at a certain tilted angle (from the OOP axis) will cause different amount of magnetization (of each component) tilting away from their easy-axis. This will then result in different slopes in the region before saturation of the AHE curve as a function of the magnetic field angle. In Fig. R1b, such tilting of the IP and OOP magnetization component with a field applied close to 45 degrees direction is illustrated. With the slope explained, the minimum of the slope is essentially when the amount of increasing AHE signal due to the IP moment tilting away from IP axis nearly cancels out the decreasing AHE signal due to the OOP moment tilting away from OOP axis. The angle where minimum slope occurs is clearly not the tilting angle of the OOP magnetization since it also depends on the IP magnetization component. However, this canting angle of the easy axis is a good indicator of whether the net magnetization favors OOP dominant or IP dominant.

With the canting angle defined in the manuscript explained, we can now address why all three samples showed some field-free switching but different magnitude in R_{AHE} loop. Sample-45 showed the best field-free switching R_{AHE} loop among the three samples because it had the best combination of OOP/IP magnetization and anisotropy strength inside the GdCo layer. Sample-13 showed very strong PMA with majority of the magnetization is OOP so the tilting of the OOP magnetization is most likely minimal albeit non-zero (from the 13.5° easy axis) resulting in the small amount of field-free switching. Although sample-56 exhibits an easy-axis canting that is 11 degrees more than sample-45, the composition of the IP/OOP magnetization can be drastically different. Based on the fact that the SOT switching R_{AHE} loop shows a very poor squareness at zero symmetry breaking field and improves as the symmetry breaking field increases, the OOP magnetization is weak in sample-56 but non-zero. As a result, field-free switching can be observed but at a very limited capacity because of the very weak PMA magnetization leading to small R_{AHE} loop.

We have taken the referee's suggestion and obtained a few different devices with different easy-axis canting angles shown in the following sections. We have made the following modification in the main text to clarify this:

- On the end of third paragraph in **Results** in page 9: *"It is to be noted that the canting angle because there is very minimal IP magnetization component in these systems."*

Comment

2. As mentioned above, the anisotropy canting angle of $\text{Gd}_x\text{Co}_{100-x}$ structures is crucial for field-free SOT switching and PMA. If the canting angle could be tuned easily, we can find the best canting angle to obtain proper PMA, field-free switching and other functions like multistate switching and ratchet effect. Therefore, the key is if the author could tune the anisotropy canting angle and obtain specific canting angle. If the canting angle can be tuned, the methods and results need to be stated more clearly.

Response

The methodology of inducing the canting in a controllable manner is an important first step to fabricating SOT devices that show interesting switching and transport phenomena. We provide a starting point that shows tunability to some extent which can be improved and optimized for specific angle resolution in future studies.

The tuning methodology we describe here involves light oxidation. We treat the fabricated devices to a very mild oxygen plasma process equivalent to a photoresist descum in nanofabrication for a set amount of time and then allowing the oxygen to migrate or diffuse into the film over a period of time. Through this oxygen plasma treatment and migration of oxygen within the film, the magnetic anisotropy can be modified which is determined by the field sweep AHE resistance measurement.

In the following paragraphs, I will describe how the magnetic anisotropy canting can be induced in detail. We start by depositing GdCo thin film heterostructure at the atomic concentration close to the magnetic compensation at room temperature which is characterized by the minimal net magnetization via magnetometry (Fig. R2a). We note that the atomic concentration we aimed for is $\text{Gd}_{21.82}\text{Co}_{78.18}$ which is slightly away from the theoretical value of 23.5% Gd. This is most likely due to the slight change in the actual growth condition or change in the distribution of atoms at a thin thickness regime (~ 10 nm). The two-phase switching can also be clearly observed here in this GdCo magnetic heterostructure close to the compensation point. From the magnetometry, the dominant magnetic anisotropy direction is IP characterized by the much stronger remanence as compared to the OOP direction.

We then fabricated Hall bar devices from the blanket thin film for transport measurements. We utilize the AHE as the signature for characterizing the canting since AHE is only sensitive to the OOP magnetization. With the varying amount of canting, the shape of the AHE field loop will be different. To induce the magnetic canting, we expose the fabricated devices to a mild oxygen plasma (50 W, 180 mTorr, 50°C chuck) for a certain amount of time. Next, we measure the AHE resistance of the device with an OOP magnetic field applied hysterically. Depending on the shape of the AHE loop, we can determine whether the perpendicular magnetic anisotropy has increased thus the magnetic canting. We repeat this process for several tries until the shape of the AHE loop no longer changes. The time between successive cycles of such oxidation and AHE measurement is about 2 to 3 hours. Next, we let the sample sit in ambient for hours and even days. This is primarily for the oxygen to diffuse or migrate from the top of the film into the bulk of the GdCo layer and even possibly reaching the top of the Ta underlayer. After letting the oxygen diffuse into the bulk of the GdCo layer, the perpendicular magnetic anisotropy is enhanced.

Figure R2: (a) Magnetometry data on Sample 1, which is deposited near the magnetic compensation composition of GdCo sandwiched between 8 nm Ta underlayer and 2 nm Pt capping. The data was obtained by VSM, probing in both the in-plane and out-of-plane direction. (b) Anomalous Hall effect resistance loop measured after a certain amount of oxidation duration and oxygen diffusion time in Hall bar device fabricated from the Sample 1 thin film.

In Fig. R2b, the AHE field loop measured after a certain duration of oxygen plasma is shown. The fabricated device possesses IP magnetic anisotropy characterized by the straight sloped line with the label “as fabricated” in light pink. As the amount of oxygen plasma exposure increases, the shape of the AHE loop begins to develop a curvature and a small hysteresis opening. After 230 s of oxygen plasma, the net magnetization can be saturated by the anisotropy field at around 200 mT which is within the largest magnetic field capability of our tool and the opening of the hysteresis is accompanied with a non-zero albeit small remanence in the OOP direction. By continuing the oxygen plasma treatment to 290 s, the anisotropy field decreases to around 100 mT. If we continue the process for another 90 s to 380 s, the overall shape and the anisotropy field remains the same (light green curve with triangle marker pointing left). If we let the sample sit in ambient for a few days (19 days for the sample (sample 1) shown in Fig. R2b), the majority of its magnetization becomes OOP. This is observable in the sharp switching of the AHE resistance with apparent coercive fields (curve with 380s Ox + 17 days). Through the angle-dependent AHE field sweep measurement, the canting of the easy-axis is 31.5 degrees from the OOP axis (Fig. R3).

To have a much better understanding of what the critical step to induce the OOP magnetic anisotropy between the oxygen injection and the oxygen diffusion is, we conducted a much more detailed experiment with shorter duration between successive steps near the magnetic anisotropy transition on another sample (sample 2) with its magnetometry result shown in Fig. R4a. In Fig. R4b, the anisotropy begins to transition away from the IP direction at 230 s of oxygen plasma. By letting the sample stay in ambient for 17 hours, the OOP anisotropy is greatly enhanced, characterized by the increase of remanence (light blue curve with triangle marker pointing up). However, by injecting more oxygen plasma for another 20 s made the OOP anisotropy worse (orange curve with square marker). By letting the sample stay in ambient for an extended period (7, 19, 25 hours), the OOP magnetic anisotropy is again enhanced with the squareness of the loop improving. For a very long period of time (58 days), the OOP magnetic anisotropy becomes very strong with almost perfect squares. From

Figure R3: Slope of the field region before fully saturating the entire magnetization in the external field direction as a function of external field angle away from the out-of-plane direction. From this measurement, 31.5° canting is found after the device fabricated from sample 1 has gone through 380 s of oxidation and 17 days of diffusion under ambient condition.

this set of experiments, it is shown that to obtain PMA in GdCo heterostructure, one only needs to inject a critical amount of oxygen that weakens the IP anisotropy and induces a small amount of OOP anisotropy. Next, the crucial step is to let the sample stay in ambient for a certain amount of time which we think is for the oxygen to diffuse into the bulk of the GdCo layer since Gd tends to be strongly oxidized. The canting angle of the easy-axis after 230 s of oxidation and 25 hours of diffusion is 22.5 degrees and after 58 days of diffusion is 13.5 degrees (fig. R5).

Figure R4: (a) Magnetometry data obtained from thin film sample 2 deposited near the magnetic compensation. VSM measurement is carried out in both in-plane and out-of-plane direction at room temperature. (b) Anomalous Hall effect resistance loop measured after certain amount of oxidation duration and oxygen diffusion time near the transition of in-plane and out-of-plane anisotropy domination. Data is taken on device fabrication from the thin film of Sample 2.

Figure R5: Canting angle of the easy-axis in the device fabricated from sample 2 with 230 s of oxidation after (a) 25 hours and (b) 58 days of oxygen diffusion in ambient environment.

At last, we would like to discuss the repeatability of such method and its controllability. We do realize that having to wait for hours or even days may not be the best way to enable industrial mass production of devices. However, the purpose of this paper is to demonstrate a method to consistently induce a canting of magnetization in GdCo rather than optimization of process development. In Fig. R2 and Fig. R4, the development of the OOP anisotropy from a perfectly IP magnetization is shown in two separate thin film samples. The fact that both samples were able to induce a strong PMA via such a method shows the consistency and accurate controllability.

We have included the following modification in the manuscript to reflect such changes:

- Fourth paragraph in **Results** on page 10: *"Recipe for inducing large canting in GdCo ferrimagnet. The canting.....should be obtainable with reasonable optimization."*
- Supplementary Note 2 for detailed description of the oxidation and diffusion process in inducing the canting.
- New main figure 2.
- New methods section: *"Oxygen plasma treatment and diffusion for inducing magnetic canting"*

Referee 02

Comment

This work investigated the SOT effect of ferrimagnetic GdCo alloys from the perspectives of field-free, anisotropy, and memristivity. The observed phenomenon is novel but arguable. I believe the manuscript remains serious problems as follows.

Response

We appreciate that the referee found our work novel and provide invaluable comments and questions to improve the manuscript.

Comment

1. I believe the most controversial point of this manuscript has to do with alloy composition. In Table I, how come sample-45 and -56 with identical composition exhibit dramatically different properties? Also, how come a minor change of the composition from Co75.3 (sample-45 and -56) to Co75 (sample-13) would result in a huge difference in magnetic properties? The readers feel confused while linking Fig. 4, 5, and 6 with Table I. The authors claimed that this is due to the compensation point where magnetic competition occurred within the bulk ferromagnet, but my argument is whether the observed goodness (field-free, memristivity, etc) is highly uncertain and uncontrollable? Note that any kind of magnetic competition, whether generated by interface exchange interaction or bulk sublattice, would create some degree of magnetic canting and instability, which could be the source of the phenomenon observed by the authors; i.e., the applied current breaks the symmetry of such kind of magnetic competition? Besides, even the Joule heating effect could disturb the spin competition and temporarily reverse the magnetic dominance from Gd and Co. In other words, the goodness generated by such conditions is in fact very unstable and difficult to manipulate.

Response

The composition in table 1 are the as-deposited composition that is calibrated with the growth rate. However, as we pointed out in the response to comment 2 of referee 1, oxygen serves as an important role in inducing the canting and affecting the PMA. We reviewed the timeline of Sample-45 and sample-56 and found that those two samples have gone through different amount of oxidation slowly over a long period in air.

In the response to comment 2 of referee 1, we have established the controllability of such canting through the oxygen plasma treatment and diffusion. The characterization of such canting via the AHE field sweep is done through a very small current $50 \mu\text{A}$. To check whether Joule heating may affect, we have conducted the same measurement at higher current values ($500 \mu\text{A}$, 1mA) and did not observe any difference in canting angle.

As we manipulate the canting with oxygen, the SOT switching behavior also closely depends on canting. We conducted the SOT switching experiment on Sample 1 (31.5° canting) and sample 2 (13.5° canting) in Fig. R6. We can see that multistate switching and ratchet effect still exists in both samples and field-free switching is more prominent in sample 1 with more canting while sample 2 with 13.5° canting also shows field-free switching but the ΔR_{AHE} is smaller.

Comment

2. I suggest the authors provide a schematic illustration that summarizes the finding. The relationship between initial magnetization, applied field, and SOT write pulse, etc is (Fig 2,3,4) not reader-friendly and hard to conclude from a quick glance.

Response

We have added some more illustrations and visual modifications to Fig. 2, 3, 4 to aid the understanding of the data better.

Comment

3. Is it possible to retrieve information about the in-plane magnetization component during SOT switching? This considers that the in-plane magnetization component plays a crucial role in ratchet

Figure R6: Field-free spin-orbit torque switching and various transport phenomena (Multistate and ratchet effect) in GdCo heterostructure with different canting angle: (a) 31.5° (b) 13.5° .

behaviors.

Response

We have attempted to extract the information about the IP magnetization component during SOT switching by varying the in-plane symmetry breaking field direction on sample 2 (13.5° canting). For all of the SOT switching curves shown so far in the manuscript, in-plane symmetry breaking field is applied along the x-direction which is the same direction as the current direction. Here, we applied the symmetry breaking field B_y along the y-direction transverse to the current direction (Fig. R7). As expected, SOT switching is heavily suppressed since symmetry is not solely broken by B_y . Despite that B_y should not break symmetry, there exists a small amount of SOT switching for B_y that are not so strong ($< |10 \text{ mT}|$). In addition, multistate switching, field-free switching and ratchet effect are still observable which may indicate that these phenomena are not dependent on the IP magnetization direction but rather the presence of non-zero IP magnetization is sufficient.

Comment

4. Following #3, as the magnetic easy axis is tilted away from the z-axis, its in-plane component is critical to SOT switching. If it is strictly parallel to SOT write current, then it can be considered an effective H_x . However, if there is non-negligible in-plane magnetization component that is perpendicular to SOT write current, the situation can be much more complex. It reminds me of the type-T tri-layer structure reported by W. J. Kong et al., Nat Commun 10, 233 (2019); both cases yield field-free switching, but the underlying mechanisms seem to be different. Therefore, it is important to verify

Figure R7: Spin-orbit torque switching of Sample 02 after 230 s of oxidation and 58 days of oxygen diffusion with varying in-plane field applied in the y-direction which is transverse to the current direction (x). Magnetization is initialized with (a) $B_z = -100$ mT and (b) $B_z = +100$ mT.

the orientation of the IP component of the canted moment. Although the authors did not mention it in the main text, it seems that they considered that the IP magnetization component is parallel to SOT write current. Also, it would be interesting to investigate the in-plane angular dependence of SOT switching by varying the relative orientation of electrical current with respect to the in-plane magnetization component.

Response

We appreciate the referee for bringing up a relevant work (W. J. Kong et al., Nat Commun 10, 233 (2019)). This work has helped us compare our system and observations to their conclusions. Our system is similar to theirs in the sense that in the GdCo layer there exists both IP and OOP magnetization components. However, in their system, both components are separated by a thin Ta layer whereas for GdCo, both components are physically connected in some way. Based on the B_y -dependent SOT switching results above in Fig. R7, the IP component is most likely parallel to the current direction during switching with the type-Z switching scenario. In W. J. Kong et al., regardless of the direction and strength (up to 30 mT) of the symmetry breaking field, SOT switching can be observed. However, for our case, when symmetry breaking field is in the y-direction almost no SOT switching is observable when B_y is larger than 10 mT. If the IP component is transverse to the current direction during switching, according to W.J. Kong et al., strong SOT switching should still be observable. However, the limited yet detectable SOT switching observed in the B_y -dependent measurement suggests that there

might be a portion of the IP magnetization with anisotropy along the y-direction, albeit remaining small.

To address this we included the following modifications to the manuscript:

- Final paragraph on page 27: ” *Since the IP magnetization is crucial the y-direction.*”
- Supplementary Figure 8

Comment

5. In the studied structure (substrate/Ta/GdCo/Pt), I see three possible sources that generate SOT: magnetization-independent SHE from Ta and Pt, magnetization-independent SHE from GdCo, and magnetization-dependent SHE from GdCo (or anomalous spin Hall effect). There might be certain SOT components that are actually dependent on magnetization, quantitative estimation of effective torque under different magnetization orientations will give more insight. Some observed phenomena may be the combined effect of magnetization-dependent SOT, tilted anisotropy, and exchange spring effects. The authors should clarify them.

Response

We understand that exploring the origin of the SOT in Ta/GdCo/Pt is an interesting subject by itself. However, this is not a critical point for the present work. We note that if the GdCo in Ta/GdCo/Pt stack has no canting, no field free switching can be observed. This shows that conventional symmetry conditions as observed in prototypical SHE stacks hold for the stack under study as well. Moreover, the quantitative estimation of the effective torques may require pinning the magnetization in various directions which makes magnetization direction dependent SOT characterization challenging.

We acknowledge that magnetization-independent SHE from GdCo can play a crucial role in SOT switching but remains as a factor only to critical switching current density characterization in addition to the contribution from adjacent spin sources such as Ta and Pt. Critical switching current density is not the main point of this manuscript. As a result, we think this question, while interesting, is not within the main scope of our study.

Comment

6. I don't see supplementary note 1 where the authors are supposed to explain the Gd symmetric 4f filling. This is important as for a symmetrically filled 4f orbital there should exist no magnetic anisotropy, but this is not the case in this work. Please explain.

Response

We apologize for missing this section. We have attached it to the supplementary file.

Referee 03

Comment

The authors have presented a detailed study of CoGd ferrimagnetic system, emphasizing on three main behaviour - field free switching, ratchet behaviour and multistate switching. The authors have

done an excellent job in convincingly demonstrating the above three behaviour.

Response

We thank the referee for finding our technical content convincing!

Comment

However, due to lack of novelty in terms of the SOT system presented and in terms of observed behaviour, I cannot recommend the manuscript for publication in Nature Communications. Field free switching has been observed and reported in ferrimagnetic system earlier (Zheng et al. Nature comm). In fact this field free switching has been observed in single layer ferrimagnetic system without any SOC layer. Similarly, multi level switching been observed in many ferromagnetic system. A similar observation in ferrimagnetic system seems to be an obvious extension and does not entail much novelty.

Response

We would like to highlight the fact that the novelty of our work lies in the observation of many different switching phenomena in a single canted GdCo system, analogous to an AFM/FM system. We also present a new way of inducing magnetic canting and enhancing PMA in such ferrimagnetic system. This method may potentially be useful for many spintronics studies that require PMA property in rare earth-transition metal ferrimagnets. We hope this clarification will delineate the uniqueness of our work in comparison to existing reports in literature.

Comment

Beside the above, I would request the author to include the recipe or work towards engineering the canted state. At present it seems like that random canted states are obtained during deposition.

Response

Please see the response to comment 2 of referee 1. We have included the recipe to induce canting state in the main text and methods section.

Reviewers' Comments:

Reviewer #1:

Remarks to the Author:

The submission has been greatly improved and is worthy of publication.

Reviewer #2:

Remarks to the Author:

This work investigated the SOT effect of ferrimagnetic GdCo alloys from the views of field-free, anisotropy, multistate, and memristivity. I agree that the observed phenomenon is novel and very interesting. From materials science point of view it does have potential for SOT-related technologies such as MRAM, memristor, neuromorphic computing, etc. However, it doesn't make sense in Table 1 that how come the three samples have identical composition would have huge differences in saturation magnetization (M_s)? Note that M_s differs from H_c , canting angle, anisotropy, etc; materials with the same atomic concentration should have close M_s . The authors mention oxygen treatment would result in different canting angles. Will it affect M_s , too? If yes, I am afraid that the actual chemical state of the claimed Gd_xCo_{1-x} formula is not really what it is---the samples might have undergone different degrees of oxidation? In the section of the recipe for inducing magnetic canting, the authors mentioned the oxidation treatment with high variability. This makes it easy to suspect that the stability of the specimen itself comes from the difference in oxidation state, which directly affects M_s , anisotropy, multi-state properties, etc. Otherwise, the transport properties along with the rest of the argument are reliable.

Response to referee comments

for manuscript NCOMMS-22-52688A

We greatly appreciate the two referees taking time to review our revised manuscript and response. We are glad that the reviewers and the editor found our previous revision to be robust and provided further opportunity for the consideration of this manuscript. In this revision, we focus on addressing the issue brought up by referee 2 regarding the chemical composition of the samples with different canting angles and their respective magnetic properties in Table 01.

Brief summary of modifications to the manuscript

Changes made under this revision in the main manuscript are in green.

0.1 Chemical composition in Table 01

- **New/modified Methods section:** Revised Table 01 with new row of oxidation treatment and clarified a few column names.

Response to individual referee comments

All responses are in blue.

Referee 01

Comment

The submission has been greatly improved and is worthy of publication.

Response

We are happy to hear that Referee 1 found our revised manuscript to be sound and worthy of publication. We appreciate referee 1's time and effort to make this manuscript much better.

Referee 02

Comment

This work investigated the SOT effect of ferrimagnetic GdCo alloys from the views of field-free, anisotropy, multistate, and memristivity. I agree that the observed phenomenon is novel and very interesting. From materials science point of view it does have potential for SOT-related technologies

such as MRAM, meristor, neuromorphic computing, etc. However, it doesn't make sense in Table 1 that how come the three samples have identical composition would have huge differences in saturation magnetization (M_s)? Note that M_s differs from H_c , canting angle, anisotropy, etc; materials with the same atomic concentration should have close M_s . The authors mention oxygen treatment would result in different canting angles. Will it affect M_s , too? If yes, I am afraid that the actual chemical state of the claimed Gd_xCo_{1-x} formula is not really what it is—the samples might have undergone different degrees of oxidation? In the section of the recipe for inducing magnetic canting, the authors mentioned the oxidation treatment with high variability. This makes it easy to suspect that the stability of the specimen itself comes from the difference in oxidation state, which directly affects M_s , anisotropy, multi-state properties, etc. Otherwise, the transport properties along with the rest of the argument are reliable.

Response

We regret the poor choice of wording which led to a confusion in interpreting Table 01.

We completely agree with the referee that the actual chemical composition of the Gd_xCo_{100-x} should be different among the three different samples in table 01 and that oxygen needs to be taken into account. We revised the name of the first row to 'As Deposited Atomic Concentration,' since those values are the original concentration as deposited without any oxidation. Those values are very close to magnetic compensation at room temperature which is the starting concentration where the easy-axis can be further engineered for canting via oxidation. Indeed, as the referee suggested, oxidation will not only change M_s but also change H_c , anisotropy and other magnetic properties. We added a new row 'Relative Degree of Oxidation treatment,' to show the relative oxidation treatment amount as a guidance for the reader on the three different samples. The actual oxidation duration is subject to slight variation in device and deposition conditions so we listed a relative strength as general guidance for the audience to recreate these canting conditions.

We are glad referee 02 found the rest of our results and arguments to be robust.

Reviewers' Comments:

Reviewer #2:

Remarks to the Author:

The revised manuscript is satisfactory. I recommend its publication.

Response to referee comments

for manuscript NCOMMS-22-52688B

We appreciate the two reviewers spending time to review our revised manuscript and responses over this entire peer review process. We are glad that both reviewers and the editor found our previous revision to be robust and recommended the revised manuscript for publication.

Response to individual referee comments

All responses are in blue.

Referee 02

Comment

The revised manuscript is satisfactory. I recommend its publication.

Response

We thank referee 2's time and effort to make this manuscript much better and are glad that referee 2 recommended the revised manuscript for publication.